



# Parameterization of the responses of subarctic European vegetation to key environmental variables for ozone risk assessment

Stefanie Falk[1,4], Ane V. Vollsnes[2], Aud B. Eriksen[2], Lisa Emberson[3], Connie O'Neill[3], Frode Stordal[1], and Terje Koren Berntsen[1]

[1]Department of Geosciences, University of Oslo, Oslo, Norway
[2]Department of Biosciences, University of Oslo, Oslo, Norway
[3]Department of Environment and Geography, University of York, UK
[4]Department of Geography, Ludwig Maximilian University of Munich, Munich, Germany

**Correspondence:** Stefanie Falk (stefanie.falk@lmu.de)

**Abstract.** The unique vegetation of the subarctic region acclimatized to extremes of cold and midnight sun are likely to be at threat from the combined impacts of climate change and increasing ozone concentrations [$O_3$]. The atmospheric and climatic characteristics of the subarctic are known to lead to pronounced peak [$O_3$] in spring. To date, only a few studies assessed the response of subarctic vegetation to variations in climate and air pollution. This study looks to fill this knowledge gap by examining essential climate variables, in particular ozone, over the past few decades. We evaluate the extent to which two recent years (2018 and 2019) deviate from climatic and [$O_3$] norms and how these potentially more frequent future deviations may influence ozone damage to subarctic vegetation. We find that 2018 was an anomalously warm and bright year, particularly in spring and early summer. Higher than average [$O_3$] was observed in April/May while frequent episodes of ozone volume mixing ratios (VMRs) above $40\,\mathrm{ppb}$ occurred in June–August. These episodes are in part attributable to forest fires in the Northern Hemisphere and warmer and sunnier conditions. We apply the integrated flux-metric Phytotoxic Ozone Dose (POD) to determine ozone risk and damage to vegetation as a function of [$O_3$], environmental factors, and species-specific physiology. Our study suggests that using generic parameterizations in assessments likely leads to underestimating the risk of ozone damage in this region. We find that bespoke parameterizations of plant functional types (PFTs) for subarctic vegetation bio-types result in an ozone-induced biomass loss of $2.5$ to $17.4\,\%$. For some species, this loss is up to $6\,\%$ larger than projected from generic parameterizations. Efforts should be targeted towards accurately defining subarctic vegetations' physiological response to essential climate variables. Our method could help to improve regional and global scale biogeochemical cycling under current and future climates.

## 1 Introduction

Ground-level ozone ($O_3$) is a highly toxic pollutant known to cause damage to human health (WHO - World Health Organization, 2008) and to a variety of ecosystems around the World (Emberson, 2020). Ozone causes reduced photosynthesis, visible foliar injury, early senescence, and programmed cell death of plants (Kangaskärvi et al., 2005). Annual global yield losses of four major crops (wheat, rice, maize, and soybean) of $3 - 15\,\%$ have been reported (Ainsworth, 2017) and a suggested loss





in primary production in forestry of $7\%$ (Wittig et al., 2009; Matyssek et al., 2012) has been attributed to ozone. The crop yield loss studies indicate a threat to food security in rapidly developing countries, e.g., in East and South-East Asia (Tang

et al., 2013; Tai et al., 2014; Chuwah et al., 2015; Mills et al., 2018), where ozone concentrations related to enhanced pollutant emissions from the transport, residential, power generation, and industrial sectors have increased in the past few decades. Even though long-term (pre-1950) observations of ground-level ozone concentrations are scarce; evidence from trend analysis of those few sites suggests that since the industrial revolution, background concentrations in the northern hemisphere have at least doubled and continued to increase (Hartmann et al., 2013). Recent trend analyses for Europe indicate that a maximum

was reached in 2007 (Derwent et al., 2018), after which tropospheric background ozone started to level off or even decline (Cooper et al., 2014; Wespes et al., 2018; Gaudel et al., 2018). This can be attributed to a successful implementation of air quality regulations coupled with economic restructuring, reducing the number of episodes of peak concentrations especially in summertime (e.g., Fleming et al., 2018; Mills et al., 2018). However, changes in environmental conditions associated with climate change, such as an increase in the frequency of heatwaves, could negate the effectiveness of emission reductions con-

cerning ozone impacts and air quality (Lin et al., 2020). Tropospheric background ozone concentrations derived from satellite instruments range between $50-65\,\mathrm{ppb}$ over Fennoscandia but strongly depend on season (Cooper et al., 2014).

The complex photochemical cycles leading to the formation of $O_3$ at ground-level involve precursor gases such as carbon monoxide (CO) and hydrocarbons known as volatile organic compounds (VOCs) in the presence of nitrogen oxides ($NO_x$). Despite a relatively short average tropospheric lifetime of approximately $22\,\mathrm{days}$ (Stevenson et al., 2005; Young et al., 2013),

ozone and its precursors are subject to advection over long distances. Episodes of high ozone concentrations in the absence of local pollution sources can therefore often be attributed to the long-range transport of both ozone and its precursors. In northern Fennoscandia, episodes of elevated ozone concentrations in 2003 and 2006 have been traced back to ozone precursor emissions related to forest fires in southern and eastern Europe, respectively (Lindskog et al., 2007; Karlsson et al., 2013). Besides VOCs from anthropogenic sources, hydrocarbons are also emitted by vegetation itself in the form of terpenes and

monoterpenes, so-called biogenic volatile substances (BVOCs). These emissions are thought to be a response to factors, such as thermal stress, defense against herbivores, or even attraction of pollinators (Peñuelas and Llusià, 2003).

In the annual cycle in Boreal climates, a distinct maximum of ground-level ozone concentrations typically occurs in spring followed by lower concentrations throughout the summer. Several conditions unique to the Arctic are stimulating the formation of this spring peak. Major pathways of ozone and its precursors' removal from the boundary layer include dry deposition on

vegetated surfaces and photochemistry (Clifton et al., 2020). Dry deposition to bare ground or snow and ice-covered surfaces was found to be low (Helmig et al., 2007). This causes a substantial suppression of ozone removal (Monks, 2000) and leads to a build-up of ozone and its precursors during winter. Come spring precursors are photochemically activated again, while snow cover typically continues to prevail and suppress ozone dry deposition. On the other hand, very high ozone concentrations occurring over shorter periods can often be attributed to deep tropopause folding events over the Arctic during winter. These

events facilitate an intrusion of stratospheric air masses which are enriched in ozone (Škerlak et al., 2015). The spring peak is more pronounced in northern Fennoscandia than in more southern locations but shows a high interannual variability (Klingberg et al., 2009, 2019). Andersson et al. (2017) showed in a modeling study focused on Fennoscandia that the variability of





ozone concentrations in winter can be attributed mainly to changes in atmospheric background (transport) of ozone, while summertime abundance is mostly affected by emissions of precursors in the rest of Europe.

The start of the growing season (GS) in northern Fennoscandia has been shifting to earlier dates in response to the general warming trend (Menzel et al., 2006; Høgda et al., 2013; Karlsen et al., 2007, e.g.) and thereby converging with the period of the ozone spring peak. At the same time, the growing season is also becoming longer. A longer growing season is prolonging the time in which vegetation can accumulate ozone. For natural and semi-natural vegetation that is already subject to rapid climatic changes known as Arctic amplification (AMAP - Arctic Monitoring and Assessment Programme, 2012; Hartmann

et al., 2013), these factors could promote a higher potential risk for northern vegetation to ozone-induced damage in the not so distant future. Recently, Hayes et al. (2021) showed that the highest sensitivity to future $[O_3]$ variability is expected in summer when vegetation is most productive.

Ozone acts as oxidative stress to plants. Its main action is imposed through reactions occurring in the cell walls and cell membranes of mesophyll cells inside the leaves. Ozone enters the leaves through stomata, leaf pores that enable gas exchange,

allowing the entry of $CO_2$ for photosynthesis and loss of $H_2O$ vapor via the plant transpiration stream. Stomatal aperture, and hence stomatal conductance to these gases (including ozone) will vary over the day and growing season primarily to balance $CO_2$ uptake against $H_2O$ vapor loss. The higher the stomatal conductance, the higher the potential for ozone uptake. Stomatal conductance has been empirically linked to environmental factors such as air temperature $T$, photosynthetic photon flux density (PPFD), vapor pressure deficit (VPD), and soil water potential (SWP) as well as photosynthesis itself (Jarvis, 1976;

Ball et al., 1987; Emberson et al., 2000; Mills et al., 2017, e.g.). Results from open-top chamber (OTC) experiments on downy birch (*Betula pubescens*) and mountain birch (*Betula pubescens toruosa*), native to subarctic regions, indicated reductions in both biomass, in root:shoot ratio, and visible leaf damage under elevated ozone treatment ($\langle[O_3]\rangle = 36 - 54\,\mathrm{ppb}$) (Manninen et al., 2009). Though, Scots pine (*Pinus sylvestris*) is considered to be more ozone tolerant due to an absence of visible injuries (Girgždienė et al., 2009), Manninen et al. (2009) found chlorophyll:carotenoid ratio and polyamines reductions under elevated

ozone concentrations indicating susceptibility to ozone also in these species.

A substantial body of evidence exists that suggests flux-based metrics, that relate stomatal ozone uptake to vegetation damage, are biologically more relevant for risk assessments than exposure-based metrics. This is because they can account for particular species characteristics (i.e. physiology and phenology) as well as environmental conditions that can decouple the

relationship between ozone concentration, ozone uptake, and consequent damage (Emberson, 2020). These flux-based metrics are better suited to represent the actual risk to vegetation from ozone, especially in the climatically extreme parts of Europe (Simpson et al., 2007; Mills et al., 2011, 2017). Most of these previous studies have focused on the Mediterranean where soil moisture deficits are thought to be most influential in decoupling concentration from uptake. Relatively few studies have explored the situation in Northern European climates (e.g. Juran et al., 2018).

To estimate the flux-based metric, also referred to as the Phytotoxic Ozone Dose over a threshold y ($POD_y$), an estimate of the stomatal $O_3$ flux ($\Phi_{sto}$) is calculated based on the assumption that the concentration of $O_3$ at the top of the canopy represents a reasonable estimate of the concentration at the upper surface of the laminar layer for a sunlit upper canopy leaf



and environmental factors affecting stomatal conductance. $\mathrm{POD_y}$ is then calculated according to:

$$\mathrm{POD_y} = \int (\Phi_{\mathrm{sto}} - y) \cdot \mathrm{dt}, \tag{1}$$

with the hourly averaged stomatal ozone flux $\Phi_{\mathrm{sto}}$ (see Appendix Eq. (B6) and a stomatal ozone flux threshold y both given in units of $\mathrm{nmol\,O_3\,m^{-2}\,PLA\,s^{-1}}$. The flux threshold y represents the detoxification potential of the plant and is typically only exceeded during daylight hours (i.e. when global radiation is above $50\,\mathrm{W\,m^{-2}}$). $\mathrm{POD_y}$ is given in $\mathrm{mmol\,m^{-2}}$ integrated over the growing season.

The $\mathrm{POD_y}$ metric is currently used in risk assessments under the United Nations Economic Commission for Europe (UN-
ECE) Long Range Transboundary Air Pollution (LRTAP) Convention to identify those locations across Europe where vegetation is at risk from ozone. This convention aims to develop an effects-based emission reduction policy that can target those ozone precursor emissions that are most influential in causing damage. To achieve this, the concept of critical levels (CLs) has been developed and applied (Maas et al., 2016). Exceedance of the CL is used to identify those areas across Europe that would benefit from targeted Europe-wide emission reductions. Methods to estimate $\mathrm{POD_y}$ and CLs are defined by the UNECE
LRTAP Convention and described in a Mapping Manual (Mills et al., 2017). The CL is calculated by:

$$\mathrm{CL_{exeed}} = \mathrm{POD_y} - \mathrm{CL}. \tag{2}$$

An application of the Mapping Manual method was made by Mills et al. (2011) where exceedance of the CL was related to observations of ozone damage to a clover bio-monitor. This study demonstrated the improved ability of flux-based metrics (in comparison to exposure-based metrics) to identify the geographical distribution of the risk of ozone damage and also
showed that the ozone damage could extend into more northerly regions of Europe. For our study, we conducted a similar bio-monitoring study at Svanhovd, a site with agrometeorological and air pollution monitoring, to assess whether local ozone concentrations were capable of inducing damage to vegetation growing under northern Fennoscandia conditions. In 2018, we indeed observed ozone damage on clover and tobacco. In contrast, no such damage was found on the clovers in 2019. This indicates that environmental factors in 2018 differed from 2019.

According to a report by the Norwegian Meteorological institute (Gangstø Skaland et al., 2019), the summer of 2018 was the warmest and driest ever recorded in eastern, western, and southern Norway. In the north (including Finnmark), it was amongst the warmest on record – favorable conditions for ground-level ozone formation. An unusually weak and northward shifted jet stream allowed for a persistent high-pressure system above northern Europe, including Fennoscandia which blocked the low-pressure systems for several consecutive months. In the period May – July, southern Norway experienced temperatures $4\,^\circ\mathrm{C}$
above normal and saw only about $60\,\%$ of normal precipitation. Northern Norway as a whole had close to normal precipitation, one of the causes of the high rate of forest fires in that area. Thermal stress on vegetation was exceptional not only in large parts of Fennoscandia but also in much of Europe, where the influence of the high-pressure system extended even over five months (April/May, July–September). These conditions gave rise to massive forest fires in different parts of Europe and thus an increase in ozone precursors. A total of 2079 forest fires were registered in Norway in 2018, twice as many as in the preceding
years 2016/17 (DSB, 2019), last accessed April 2020). In Sweden, about 500 fires had been reported (five times more than in a




usual summer), and an estimated total of $25000\,\mathrm{hectare}$ burned down in central Sweden (Gävleborgs, Jämtlands, and Dalarnas län) (Björklund et al., 2019). Boreal wildfires are found to emit ozone precursors CO ($[CO]/[CO_2] \propto 6-13\%$) and VOCs ($[VOCs]/[CO_2] \propto 0.5-1.5\%$) (Cofer et al., 1990). Coincident peak $[O_3]$ is found in ozone monitoring data (Fig. 2) in July. It is very likely that elevated ozone, like during the 2003 drought period (Solberg et al., 2008), was promoted by a combina-
tion of various factors such as wildfires, reduced cloud cover (increased solar radiation), reduced dry deposition and turbulent mixing due to the stagnant weather conditions, and increased BVOC emissions. As such events may become more frequent in the future, conditions at Svanhovd for the growing seasons 2018 shall serve as a reference for probable future conditions in northern Fennoscandia. Conditions in 2019 represent the present climate.

The described methods for risk assessment rely on accurate representation of plant physiological responses to key environmental variables and are most often parameterized using data for less extreme climates, i.e., temperate or continental climates; and to a lesser extent, Boreal and Mediterranean climates. Parameterization of these models for more extreme bio-geographical regions such as subarctic climates has not been performed and relies on assuming that less extreme Boreal parameterization will be applicable. It is important to understand the implications of using these more generic parameterizations for more ex-
treme climate regions. In our study, we explore how risk might be over- or underestimated using generic vs bespoke subarctic parameterization. We also investigate the effects of the interannual variability of essential climate variables (sensu World Meteorological Organization) on future risk of vegetation damage. This allows an understanding of how the threat from ozone may change in the future and which are the key aspects of plant physiology that might determine the potential risk. Hence, we determine which parameters require special attention for more accurate pollution impact assessment and more generally
for describing gas exchange response to a changing climate that will influence biogeochemical cycling. This might also be one key to solve long-standing issues of earth system models (ESM) which often lack PFTs specifically representing subarctic conditions in their land surface models (Poulter et al., 2015; Lawrence et al., 2019).

The observation site Svanhovd and available observational data are presented in Section 2. From these data, we derive multi-annual averages of essential climate variables (temperature, precipitation, global irradiance, and ozone) and evaluate the
deviation of 2018/19 from this norm (Section 3). In Section 4, we use the $DO_3SE$ model to estimate the ozone uptake by natural and semi-natural vegetation to investigate the main environmental drivers for projected ozone damage risk in 2018. To this end, we derive bespoke parameterizations for subarctic species to assess over- and underestimations arising from the use of generic parameterizations established from less climatically extreme conditions. We will summarize our results in Section 5 and discuss how this research can support improvements in future assessments of pollution risk and more generally biogeochemical
cycling.

## 2 Data acquisition

We performed ozone concentration measurements in the subarctic, during the growing seasons of 2018 and 2019. Our chosen location was Svanhovd ($69.45\,^{\circ}$N, $30.03\,^{\circ}$E, $30\,\mathrm{m\,a.s.l.}$) for several reasons: (1) Ozone had been monitored there in the past,





(2) an agrometeorological station is measuring, e.g. 2 m temperature, precipitation, global irradiance, and soil temperature at
different depths, and (3) a national surveillance of air pollution ($NO_x$, $SO_x$, PM, radiological species) is ongoing. Most of these
measurements are performed by the Norwegian Institute for Air Research (NILU), who also conducted ozone measurements
on our behalf. In addition, we collected atmospheric monitoring and ozone monitoring data taken at Svanhovd over the past
35 years.

All measured essential climate variables for the 2018/19 growing season were obtained through luftkvalitet.no operated by
NILU (a) (last accessed May 2020). Our ozone monitoring data have been added to luftkvalitet.no and the EBAS database
operated by NILU (b) and are thus openly accessible. Long-term ozone observation data (1986–1996) were obtained from
EBAS. Note that data from before the 1990s do not follow the high-quality standard procedures implemented nowadays and
have to be treated with care (Solberg, 2003). Especially, ozone monitors did not undergo regular re-calibration. That probably
led to drifts in the observed data and may impose a false trend which shall not be of concern for us as we are interested in the
seasonality. The long-term data sets of agrometeorological variables including temperature and precipitation are available from
September 1992 to the present day (LandbruksMeteorologiske Tjeneste NIBIO, note the station name here is Pasvik).

To relate local ozone concentration [$O_3$] with visible damage on ozone sensitive plants, both an ozone bio-monitor (referred
to as ozone garden) and a conventional ozone monitor using the existing atmospheric monitoring infrastructure had been
installed during the 2018/19 growing season at Svanhovd. We report [$O_3$] as volume mixing ratios (VMR) in ppb for easier
comparison with exposure-based risk assessment. The locations of the atmospheric monitoring site and the ozone garden are
marked in the aerial photography shown in Fig. 1a. An ozone garden consists of selected plant species that are sensitive to
ozone and are likely to display visible injuries. Cultivated species were, e.g. clovers (*Trifolium repens*, *Trifolium pratense*),
tobacco (*Nicotiana tabacum*, cultivars Bel-W3, Bel-B and Bel-C), and potato (*Solanum tuberosum*). As shown in Fig. 1b, the
plants had to be protected from herbivores with a wire-mesh fence. In 2018, we qualitatively observed visible ozone damage
on semi-natural vegetation (clover) and crops (tobacco). In contrast, no such visible damage was found on the clovers in 2019,
although it was observed on the sensitive tobacco cultivar (Bel-W3). This indicates that the vegetation was more affected by
ozone in 2018 than in 2019.

All relevant data for the growing season 2018/19 are shown as time series in Fig. 2. Ozone concentrations measured at
2 m height above ground are averaged hourly. The hatched areas mark times when no ozone data were recorded. Note, while
the downtime during winter was planned, missing data in two weeks of July 2018 (July 9–23) were due to problems in data
acquisition.

As can be seen in Fig. 2a, [$O_3$] peaks in spring (April/May) and reaches its minimum in late summer (July/August). The
spring peak has not been captured completely in 2019, since data acquisition only started in late April. In summer 2018
(June–August), high ozone concentrations ([$O_3$] > 40 ppb) were recorded 50 times. The highest summer ozone concentration
([$O_3$] = 50.2 ppb) was measured on July 25. This coincides with the period of the most extensive forest fires in central Sweden
which occurred from July 12–29 (Björklund et al., 2019). However, due to the above-mentioned data acquisition problems, we
missed most of the corresponding ozone data for this event and most likely also the peak [$O_3$]. A method for gap-filling these





(a)

(b)

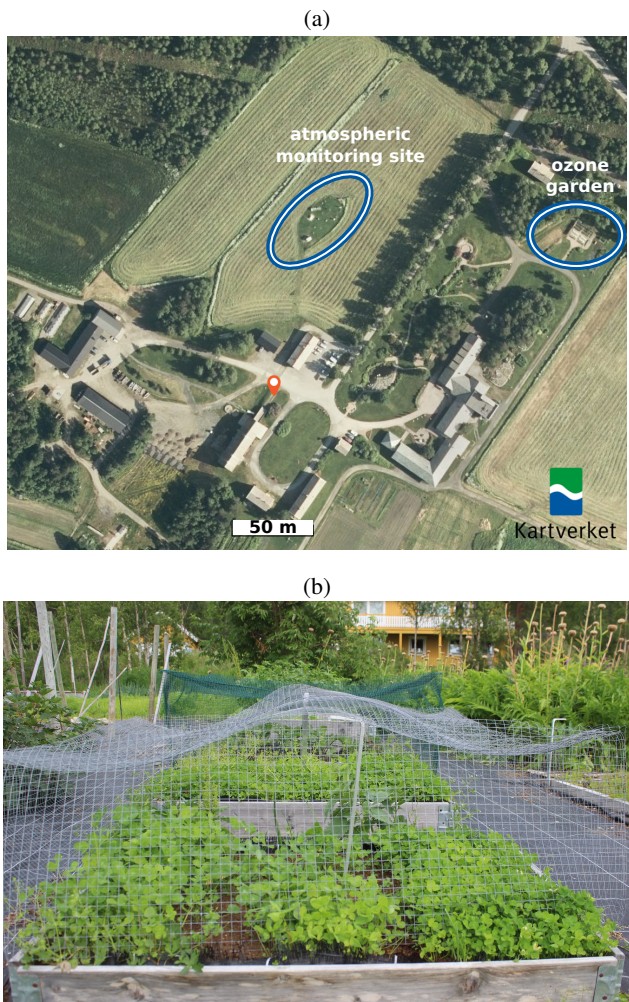

**Figure 1.** NIBIO Environment Centre Svanhovd close by the settlement of Svanvik, Norway. (a) Atmospheric monitoring site and ozone garden have been marked. Aerial photography ©Norges Kartverk; (b) Clovers in the Svanhovd ozone garden. The plants had to be secured against herbivores with a wire-mesh fence. The plants shown are approximately $6 - 12\,cm$ high.

data has been presented in Falk et al. (2021). In contrast, $[O_3]$ only rose 18 times above the threshold of $40\,ppb$ during the

summer of 2019.

Hourly averaged $2\,m$ temperatures above $20\,°C$ occurred more regularly in July 2018 than in 2019 (Fig. 2b). In 2018, spring temperature regularly rose above freezing only in May, while in 2019 this occurred already early in March/April. More rain events with accumulated daily precipitation ($\sum_d \text{Precip}$) above $10\,mm$ occurred in the summer of 2018 compared to 2019 (Fig. 2c). Qualitatively, global irradiance ($Q_0$) displayed in Fig. 2d was higher in May and July 2018 compared to 2019, while



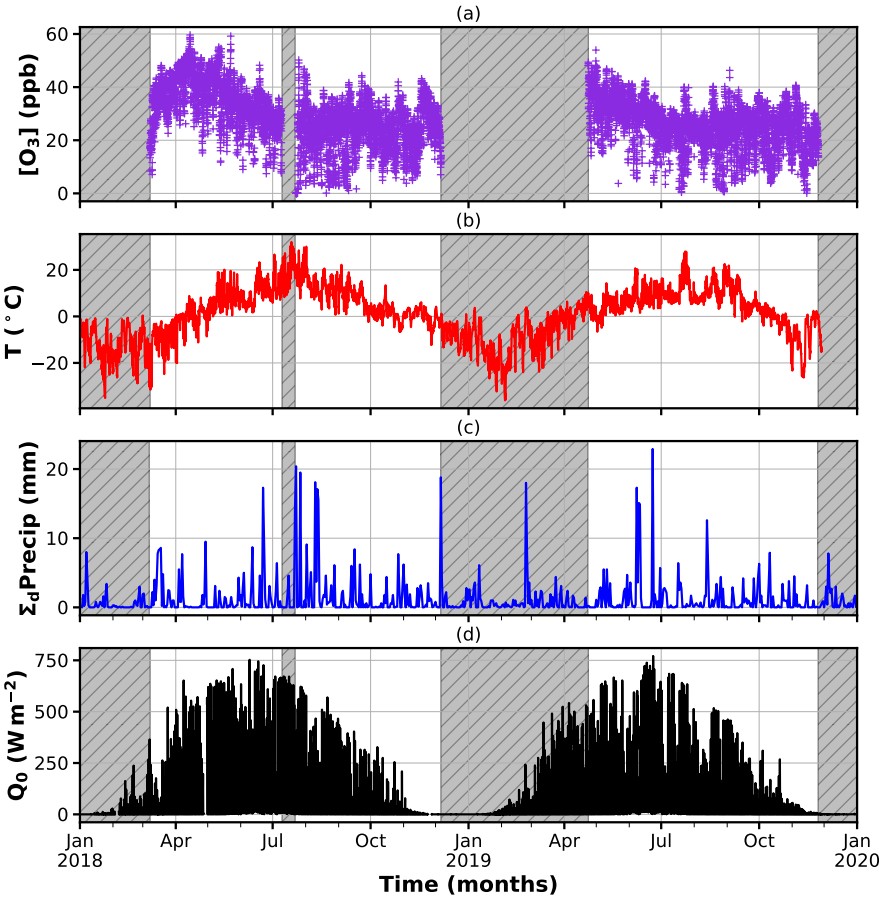

**Figure 2.** Observational data from atmospheric monitoring at Svanhovd in 2018/19. The hatched areas indicate periods without ozone monitoring data. (a) Hourly averaged $[O_3]$; (b) hourly averaged temperature; (c) daily accumulated precipitation; (d) hourly averaged global irradiance.

June 2019 showed higher irradiance than in 2018. A high global irradiance, in most cases, is the result of a low cloud fraction. In both years, the maximum recorded $Q_0$ was $750 \, \mathrm{W \, m^{-2}}$ and reached in June.

## 3  Statistical analysis

In this section, we assess the climate conditions at Svanhovd. To this end, we compute a multi-annual average (referred to as climatology) for each environmental key variable (temperature, precipitation, global irradiance) and ground-level ozone.
We evaluate the statistical significance of divergences from the norm in these variables (referred to as anomalies) in 2018/19. Concerning ozone, we performed a bias correction accounting for an increase in tropospheric background $[O_3]$ and cross-

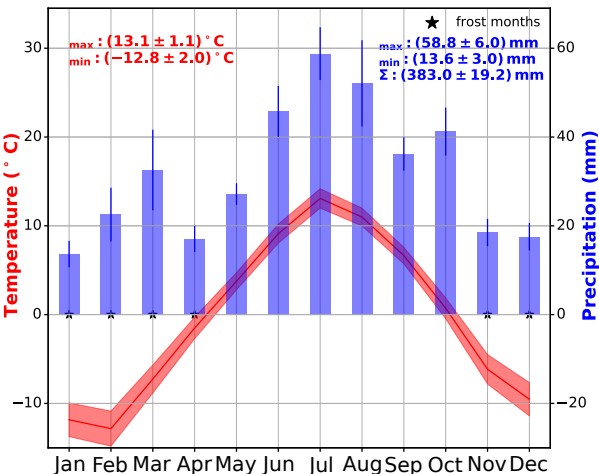

**Figure 3.** Climate diagram following Walter and Lieth. The diagram is based on climatological data for Svanvik/Pasvik (1992–2012). Monthly averaged temperatures (red line) are displayed with standard error of the mean as error band. Averaged monthly accumulated precipitation (blue bars) is shown with a standard deviation of mean as error bars. Months with average temperatures below freezing are denoted with a star.

calibrated the climatology with long-term observations at other sites in northern Fennoscandia as described in Falk et al. (2021).

### 3.1 Derived climatologies

Svanhovd is located within the subarctic climate zone. Climatologies of temperature and precipitation derived for 1992–2012 are shown in a combined climate diagram (Fig. 3). Monthly averaged $2\,\mathrm{m}$ temperatures (red line) are displayed with a standard error of the mean (SE) as error band. We chose SE in this case due to the large interannual variability in temperature reflected in the standard deviation. Averaged monthly accumulated precipitation (blue bars) is shown with standard deviation as error bars. Months with average temperatures below freezing are denoted with a star. As can be seen, temperatures stay below freezing

for 5 consecutive months, while only 2 months breach $10\,^{\circ}\mathrm{C}$ regularly (July, August), satisfying the conditions for Köppen's climate classification of a regular subarctic climate (Dfc) (e.g. Beck et al., 2018). The highest monthly average temperature is $(13.1 \pm 1.1)\,^{\circ}\mathrm{C}$ in July and the lowest $(-12.8 \pm 2.0)\,^{\circ}\mathrm{C}$ in February. The coldest measured temperature was $-45.2\,^{\circ}\mathrm{C}$ in January 27, 1999, while the highest temperature $(29.4\,^{\circ}\mathrm{C})$ occurred on July 16 the same year.

The average accumulated monthly precipitation ($\sum_d \mathrm{Precip}$) indicates that winter and spring (November–April, except for

March) are relatively dry ($\sum_d \mathrm{Precip} < 20\,\mathrm{mm}$). March precipitation is primarily snow and will influence the start of the growing season. The driest month is January with $\sum_d \mathrm{Precip} = (16.7 \pm 3.0)\,\mathrm{mm}$. The most precipitation occurs in the summer months, with a $\sum_d \mathrm{Precip} = (58.5 \pm 9.2)\,\mathrm{mm}$ in August. The average annual accumulated precipitation ($\sum_m \mathrm{Precip}$) given with standard error of mean is $(383 \pm 86)\,\mathrm{mm}$. Precipitation shows a considerable interannual variability.





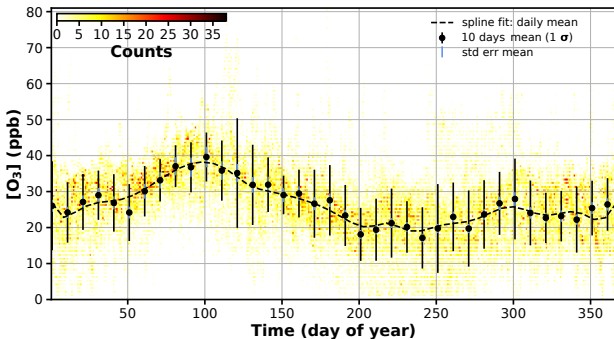

**Figure 4.** Svanhovd ozone climatology. The density distribution is shown together with a multi-annual average of daily $[O_3]$ (climatology). A spline has been fitted through daily mean $[O_3]$. The climatology of daily mean ozone is shown together with $1\sigma$ standard deviation and standard error of the mean.

The climatology for global irradiance ($Q_0$, not shown) approaches zero in December/January (polar night) and reaches its maximum in June/July (midnight sun conditions). We deduce the corresponding highest $Q_0$ to $800\,\mathrm{W\,m^{-2}}$. The maximum of the lowest $Q_0$ (including night) amounts to $200\,\mathrm{W\,m^{-2}}$ indicating that a considerable amount of light is available under midnight sun conditions.

The daily mean ozone climatology ($\langle[O_3]\rangle$) (1986–1996) is shown in Figure 4. Darker colors indicate higher probability to observe these values. On top of the density distributions, a $10\,\mathrm{days}$ average of daily mean ($\langle[O_3]\rangle_{10\,\mathrm{d}}$) is displayed together with $1\sigma$ uncertainties and SE, respectively. A spline was fitted through the data to guide the eye. The $[O_3]$ density distribution and the $\langle[O_3]\rangle_{10\,\mathrm{d}}$ display the expected, pronounced seasonal cycle. The spring peak occurs on average on day of the year (doy) 100 and amounts to $40\,\mathrm{ppb}$ while the annual average $\langle[O_3]\rangle$ is $28\,\mathrm{ppb}$. The decline in $\langle[O_3]\rangle$ coincides with the average beginning of $CO_2$ uptake by coniferous trees (Kolari et al., 2007; Wallin et al., 2013). In July–September (doy 182–273), ozone is occasionally almost completely depleted. This hints to ozone uptake by vegetation even at low light intensities during midnight sun conditions in combination with stable planetary boundary layer conditions preventing mixing of ozone rich air.

## 3.2 2018/19 anomalies

In the following, we look at the 2018/19 anomalies and discuss the differences between the two years. As weather extremes like in 2018 are likely to become more frequent with climate change, this will help to assess possible future changes in the vulnerability of subarctic vegetation to ozone.

In Fig. 5a, temperature anomalies for Svanvik are shown as a percentage of days warmer or colder than $\pm1\sigma$ from the climatology for each month. Negative deviations from climatology are displayed as a negative percentage. The annual positive/negative deviations are indicated in the respective corners (right upper/right lower). The hatched area between the dashed lines indicates the expected percentage of values falling above/below $\pm1\sigma$ if a normal distribution is assumed (15.9 %). We





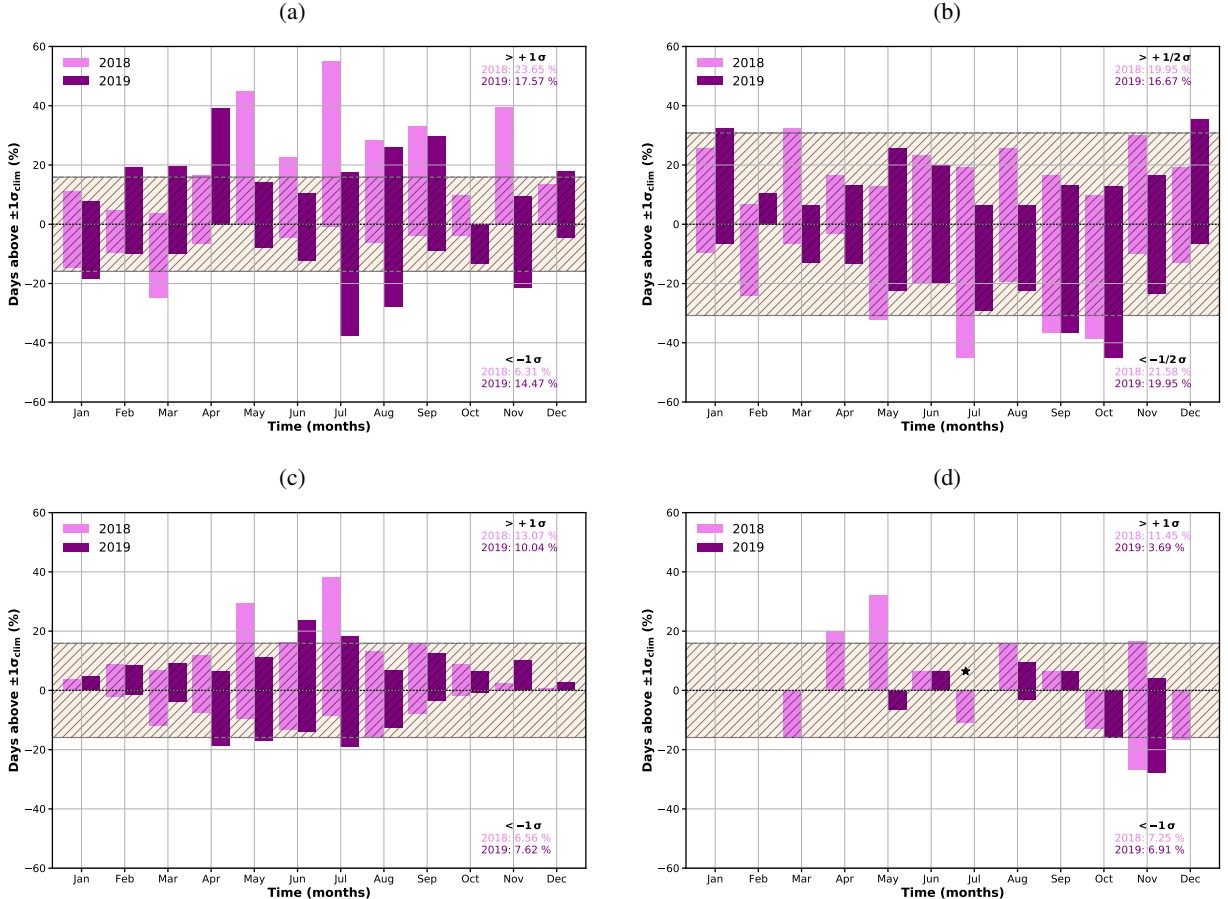

**Figure 5.** 2018/19 anomalies of key environmental variables at Svanhovd displayed as percentages of days significantly deviating from climatological mean through standard deviation for each month. Negative deviations from the climatology are shown as a negative percentage. The annual positive/negative deviations are indicated in the respective corners (right upper/right lower). The hatched area between the dashed lines indicates statistical significance under the assumption of a normal distribution ($15.9\,\%$, $30.8\,\%$ for Precip.). (a) Temperature; (b) precipitation; (c) global irradiance; (d) ozone.

find, that the summer of 2018 was significantly warmer than average. Especially in May and July more than $40\,\%$ of the days were significantly warmer. A significant number of warmer days continued to occur in all months (August–November) except for October. March 2018 had many unusually cold days. The summer of 2019 was fairly average, however, more significantly colder days occurred in July, while August had both significantly colder and warmer days. April 2019 had significantly warmer than average days.

Similarly, the precipitation anomalies for Svanvik are displayed in Fig. 5b. Due to the large interannual variability in observed precipitation, only a few days deviated significantly from the climatology on the $1\sigma$ level. Therefore, we lowered the $\sigma$ level constraint to $\pm\frac{1}{2}\sigma$. The percentage of days wetter/drier than $\pm\frac{1}{2}\sigma$ are shown for each month. Positive deviation refers to days





wetter than average and negative deviation to drier than average. The hatched area between the dashed lines indicates the expected percentage of values falling above/below $\pm\frac{1}{2}\sigma$ if a normal distribution is assumed (30.8 %). Unlike the temperature

anomalies, the picture is not as clear. While March 2018 had a significant number of days wetter than usual, summer and fall (May/July, September/October) had a significant number of days drier than average. 2019 was rather average throughout the year, but with a significant number of days drier than normal in September/October and wetter in December.

The global irradiance anomalies for Svanvik are presented in Fig. 5c. The percentage of days above or below $\pm 1\sigma$ from the climatological mean is displayed for each month. In summer 2018 (May/July), global irradiance showed a significant number

of days with higher than average irradiance, while 2019 was generally an average year with a significant number of days darker in spring (April/May).

We use the bias-corrected and cross-calibrated ozone climatology (Falk et al., 2021) and assess the monthly significance of the ozone concentration anomalies in 2018/19. The resulting percentage of days above/below $\pm 1\sigma$ is displayed for each month in Fig. 5d. Note that missing bars for months in winter and spring are both due to the temporal extent of our data and anomalies

not meeting the $1\sigma$ criteria. The star indicates the reconstructed data in July. In April/May 2018, ozone was significantly elevated, while no significant enhancement is found in 2019. In both years, ozone concentrations were significantly lower than average in November. These results suggest that stagnant meteorological conditions in April/May in combination with a late start of the growing season contributed more to elevated ozone than the extensive forest fires in Sweden that did not significantly enhance $[O_3]$ in 2018.

In summary, 2018 had a significant number of days that were warmer, drier, and brighter than the climatology, while 2019 was a rather average year. Ozone was not significantly enhanced during the GS 2018/19, but a substantially higher number of peak $[O_3]$ were observed in 2018 than in 2019.

## 4   DO3SE modeling

We investigate the difference between 2018 and 2019 concerning stomatal ozone uptake and assess which of the environmental

factors might be most influential in determining $O_3$ damage risk. We model ozone uptake of natural and semi-natural vegetation using the $DO_3SE$ model (Büker et al., 2012) and develop bespoke parameterizations for common plant species in our focus area. First, we give an account of the methodology for deriving bespoke parameters, after which we present and discuss the modeling results. The relevant components of the $DO_3SE$ model are summarized in Appendix B.

### 4.1   Model parameters

We deduced the dominant species on-site at Svanhovd and from Fig. 1 and found perennial grassland, birch (generalized as deciduous trees), and Scots pine (generalized as coniferous trees). Generic parameters for these PFTs are derived from Simpson et al. (2007); Mills et al. (2011, 2017) and will be referred to as mapping manual (MM) parameterizations in the following. Our parameterization of coniferous trees is based on the MM's Boreal Norway spruce, deciduous trees on the MM's silver





birch, and perennial grassland on MM's perennial grassland for central Europe. For a comprehensive list of generic MM model
parameters, consult Supplement Table S1.

Initial test simulations with the MM PFTs revealed an unrealistically low stomatal conductance at leaf-level ($G_{\mathrm{sto}}^{\mathrm{leaf}}$) in 2019
particularly for perennial grassland (Appendix Fig. B1). Substantial stomatal conductance in this species occurred only during
an extended warm period in late July. In ecological terms, this means that there was almost no growth of grass in the summer
of 2019 – a prediction that is easily falsified by reality. We identified $f_{\mathrm{T}}$ as abnormally low, being the limiting factor of
stomatal conductance in perennial grassland. Therefore, we saw the necessity to propose bespoke parameterizations describing
an acclimation of the generic PFTs to the climatic conditions in the focus area. Note, however, that these parameterizations are
hypothetical and have yet to be verified by experiments.

We presume that perennial vegetation will likely maximize carbon acquisition during the short subarctic growing season.
Because the MM version of the $DO_3SE$ model currently does not simulate net photosynthesis ($A_{\mathrm{n}}$), we assume a first order
proportionality between $A_{\mathrm{n}}$ and $g_{\mathrm{sto}}$ (Medlyn et al., 2011) and effectively tune the temperature and light response functions ($f_{\mathrm{T}}$
and $f_{\mathrm{light}}$) for higher $\langle g_{\mathrm{sto}} \rangle / g_{\mathrm{max}}$ to acclimate the growth potential to the local climate. For details regarding these functions
see Appendix B. Further, we presume that $f_{\mathrm{VPD}}$ and $f_{\mathrm{SWP}}$ suit our vegetation types and no acclimation is necessary for these.

We define two different hypothetical temperature acclimations: subarctic and cold. We construct *cold* as representative for
a species that is more tolerant to cold temperatures, but slightly less efficient at warm temperatures compared to MM. This
is accomplished by moving $T_{\mathrm{opt}}$ towards cooler temperatures while keeping the other parameters fixed to MM values. In the
same way, *subarctic* is constructed to represent a species that is very tolerant to cold but sensitive to high temperatures, and
most efficient at cool temperatures. Therefore, we shift $T\mathrm{min}$ and $T\mathrm{max}$ to colder temperatures and choose $T_{\mathrm{opt}}$ close to the
climatological mean temperature of 1992–2000. In other words, we aim to increase the overlapping area between $f_{\mathrm{temp}}$ and the
temperature probability density function (PDF) during the GS. The most extreme of the two acclimation types is *subarctic*. In
Fig. 6a, we show the resulting $f_{\mathrm{temp}}$ for perennial grassland together with temperature PDFs deduced from the five months GS
in two climatological periods. The periods 1990s and 2000s have been chosen to indicate that the vegetation has been subject
to climate change. Thus, the predominant acclimation at present is not known.

The resulting response functions for coniferous and deciduous trees are displayed in Appendix Figs. B3a and B4a, respec-
tively. We base our temperature acclimation of coniferous trees on experimental results on Norway spruce which were found
to be active already at rather low air temperatures and can reach $60\%$ photosynthetic activity as early as $\mathrm{doy}$ 100 (Kolari
et al., 2007; Wallin et al., 2013). In particular, we use the time series of $CO_2$ uptake and temperature at observation sites in
southern ($\rightarrow$ *cold*) and northern ($\rightarrow$ *subarctic*) Finland by Kolari et al. (2007) to estimate the optimal temperature interval
($10\,^{\circ}\mathrm{C} \leq T_{\mathrm{opt}} \leq 15\,^{\circ}\mathrm{C}$). We presume a similar temperature acclimation for deciduous trees. All bespoke temperature parame-
ters are listed in Table 1.


Regarding the acclimation of $f_{\mathrm{light}}$, we presume that the opening extent of stomata at low light conditions differs in subarctic
species compared to species in less extreme climates. We, therefore, adjust the light sensitivity of stomata such that the PPFD
value needed to reach $50\%$ opening is varied $\pm 20\%$. For this we derive the inverse function $f_{\mathrm{light},k}^{-1}$ of Eq. (B4) analytically



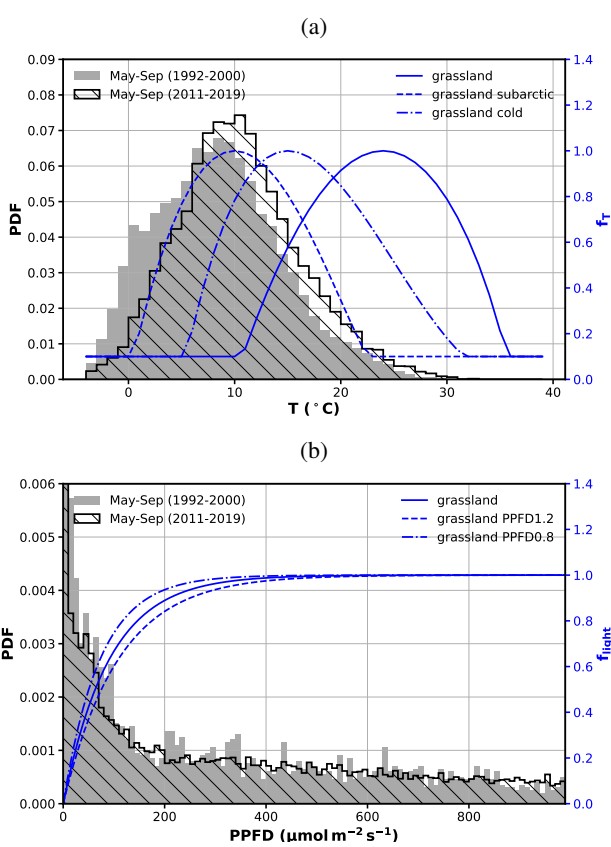

**Figure 6.** Construction of bespoke response functions for grassland. (a) $f_{\mathrm{T}}$ and (b) $f_{\mathrm{flight}}$ are shown together with underlying $T_{\mathrm{air}}$ and $Q_0$ climatologies (probability density function - PDF), respectively. Original mapping manual parameterization is shown in comparison as solid line. Note that $Q_0$ has been truncated to 0.006. PPFD0.8 and PPFD1.2 refer to $\alpha$ values increasing/decreasing PPFD at $f_{\mathrm{light}} = 0.5$ by $\pm 20\,\%$, respectively.

for each PFT or species $k$

$$\gamma_k(f_{\mathrm{light}}) := f_{\mathrm{light},k}^{-1} = -\frac{\ln(1 - f_{\mathrm{light}})}{\alpha_k}. \tag{3}$$

First, we solve Eq. (3) for the MM default value $\alpha_{\mathrm{MM}}^k$ at 0.5

$$\gamma^k(0.5) = -\frac{\ln(0.5)}{\alpha_{\mathrm{MM}}^k}, \tag{4}$$

and define a variation $\gamma' = \gamma \cdot \eta$ with $\eta \in \{0.8, 1.2\}$. By inverting Eq. (3) for $\alpha$, we find $\alpha'(\gamma')$. The resulting functions for perennial grassland (PPFD0.8, PPFD1.2) are shown in Fig. 6b) (for deciduous and coniferous trees refer Appendix Figs. B3b and B4b. All derived parameters are tabulated in Table 2.

As indicated above, a better acclimation to prevailing climate conditions is characterized by a higher average $\langle g_{\mathrm{sto}} \rangle / g_{\mathrm{max}}$. To compare our bespoke parameterizations, we propose the following metric. We calculate an average relative stomatal con-





**Table 1.** Bespoke temperature parameterizations. MM refers to mapping manual (Mills et al., 2011, 2017).

| Species | type | $T_{\min}$ | $T_{\mathrm{opt}}$ | $T_{\max}$ |
|---|---|---|---|---|
| | MM | 5 | 20 | 100 |
| Deciduous tree | Cold | 5 | 15 | 100 |
| | Subarctic | 0 | 10 | 100 |
| | MM | 0 | 20 | 100 |
| Coniferous tree | Cold | 0 | 15 | 100 |
| | Subarctic | 0 | 10 | 100 |
| | MM | 10 | 24 | 36 |
| Perennial grassland | Cold | 5 | 15 | 36 |
| | Subarctic | 0 | 10 | 24 |

**Table 2.** Bespoke light parameterizations. MM refers to mapping manual (Mills et al., 2011, 2017).

| Species | type | $\alpha$ | $\gamma(0.5)$ |
|---|---|---|---|
| | MM | 0.004 | 165.035 |
| Deciduous tree | PPFD0.8 | 0.005 | 132.028 |
| | PPFD1.2 | 0.003 | 198.042 |
| | MM | 0.006 | 115.525 |
| Coniferous tree | PPFD0.8 | 0.008 | 92.420 |
| | PPFD1.2 | 0.005 | 138.629 |
| | MM | 0.011 | 63.013 |
| Perennial grassland | PPFD0.8 | 0.014 | 50.411 |
| | PPFD1.2 | 0.009 | 75.616 |

ductance $\langle g_{\mathrm{sto}} \rangle / g_{\max}$ (Eq. (B1) with $f_{\mathrm{SWP}} = 1$) at noon ($11\,\mathrm{am} - 1\,\mathrm{pm}$ local time) and in the morning ($5 - 9\,\mathrm{am}$ local time) for the whole record of key environmental factors' data and apply mean and standard deviation. We presume that $\langle g_{\mathrm{sto}} \rangle / g_{\max}$

around noon (highest light intensity) is a good proxy for $CO_2$ uptake efficiency (growth). In addition, a small deviation between noon and morning and a low standard deviation would indicate higher robustness to variability in growing conditions. Based on these criteria, we identify *subarctic*–PPFD0.8 as the best parameterization for all PFTs (Fig. 7). As expected due to the small adjustments, coniferous trees display the smallest differences between the different parameterizations, while the differences for perennial grassland are substantial as a response to the proposed temperature acclimation.

Another important factor is the timing and length of the GS. Start ($A_{\mathrm{start}}$) and end ($A_{\mathrm{end}}$) of GS for each PFT are given in $\mathrm{doy}$ (Table 3) and estimated as follows. For coniferous trees, we use a MODIS (Aqua/Terra) retrieved net photosynthesis product



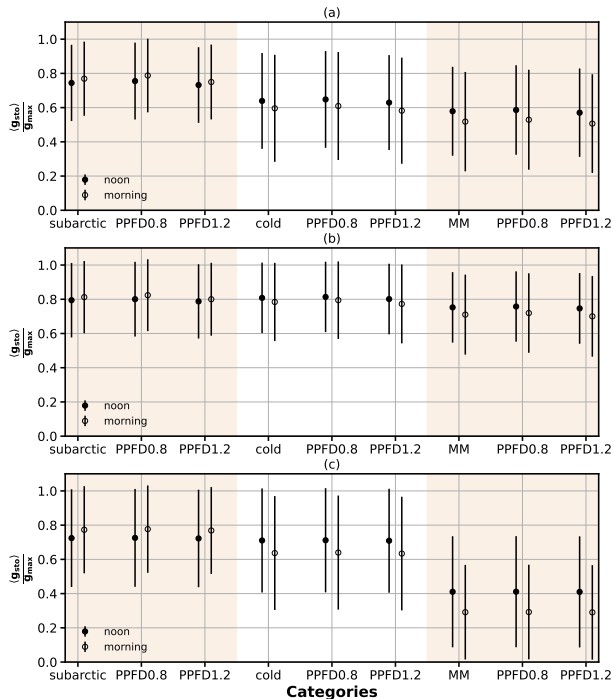

**Figure 7.** Proposed metric to test bespoke response functions. GS (May–August) climatological averages and standard deviation of $g_{sto}^k$ (Eq. (B1)) relative to $g_{max}^k$ at noon (averaged over $11\,\mathrm{am} - 1\,\mathrm{pm}$ local time) and in the morning (averaged over $5 - 9\,\mathrm{am}$ local time) are shown for (a) deciduous trees; (b) coniferous trees; (c) perennial grassland.

(Running et al., 2015) for a $1 \times 1\,\mathrm{km}$ patch centered at Svanhovd to determine $A_{start}$ and $A_{end}$. The net photosynthesis vs. time of year data can be approximated by a second-order polynomial function (Fig. B2). By calculating the roots of the fitted polynomial, we derive $A_{start}$ as doy 122 for 2018 and doy 106 in 2019. $A_{end}$ amounts to doy 261 and 274, respectively. This value will be used for all PFTs alike. The resulting growing season for coniferous trees in 2019 is, hence, one month longer than in 2018. For estimating $A_{start}$ of deciduous trees, we use the 5 consecutive days above $5\,^\circ\mathrm{C}$ agricultural rule of thumb on gridded temperature data from SeNorge.no (Appendix Fig. A1). We find doy 129 and 130, respectively. For both years, these dates coincide with the first snow-free day at the closest inland meteorological weather station at Øvre Neiden (Sør-Varanger, NOR). For comparison, the generic dates for central Europe ($A_{start} = 100$ and $A_{end} = 307$) indicating a substantially longer growing season. Due to a lack of quantitative field observation at Svanhovd, we assume a latency period of $1\,\mathrm{month}$ after snow-melt for perennial grassland. This assumption is supported by observations in Rovaniemi, Finland (Korhonen et al., 2018, Supplement Fig. S1).

Soil texture influences water availability and thus $\mathrm{POD_y}$ calculations in $\mathrm{DO_3SE}$ (Büker et al., 2012). The soil at Svanhovd is characterized in an ICP Forest plot survey as gleyic cambisols according to the Food and Agriculture Organization of the





**Table 3.** Start and end of growing season in doy.

| Species | Year | $A_{\text{start}}$ | $A_{\text{end}}$ |
|---------|------|---------|---------|
| Default[b] | – | 100 | 307 |
| Deciduous tree | 2018 | 129* | 261[a] |
| | 2019 | 130[c] | 274[a] |
| Coniferous tree | 2018 | 122[a] | 261[a] |
| | 2019 | 106[a] | 274[a] |
| Perennial grassland | 2018 | 159[d] | 261[a] |
| | 2019 | 161[d] | 274[a] |

[a] MODIS (Aqua/Terra) net photosynthesis product;

[b] Simpson et al. (2012),

[c] $5\,\text{days} - 5\,°\text{C-rule}$, $T_{\text{air}}$ from seNorge.no;

[d] One month after snowmelt; reference station Øvre Neiden
(Sør-Varanger, NOR)

United Nations (FAO) soil classification system and as eluviated dystric brunisol in the ICP Forest soil classification system (V. Timmerman, personal communication, June 2020) Within the $\text{DO}_3\text{SE}$ soil parameterization, the sandy loam texture describes the upper $60\,\text{cm}$, where the plant roots are found, of this soil type best.

    The estimate of leaf boundary layer resistance is a function of wind speed which will be influenced by the tree height and leaf dimensions. A sample of downy birch leaves collected at Svanhovd had an average length of $(3.0 \pm 0.5)\,\text{cm}$ that is smaller

than in the MM parameterization. Systad et al. (2004, p. 52) indicate an average tree height of $13.5\,\text{m}$ in the area. One Scots pine forest at Svanhovd has been studied as a part of the ICP Forest mapping. Heights were measured in 2004 when the stand was 90 years old. Mean tree height was $10.1\,\text{m}$ and maximum height was $16.2\,\text{m}$ (V. Timmerman, personal communication, June 2020). We used $13.5\,\text{m}$ height for both deciduous and coniferous trees.

### 4.2   $\text{POD}_1$ results and implications on ozone impact

We have modeled $\text{POD}_y$ with a flux threshold $y = 1\,\text{nmol}\,\text{m}^{-2}\,\text{s}^{-1}$ per projected leaf area (PLA) for the three natural/semi-natural vegetation types, deciduous trees, coniferous trees, and perennial grassland using the generic MM parameterization and our bespoke temperature (*cold*, *subarctic*) and light (PPFD0.8, PPFD1.2) parameterizations. Canopy height and leaf size are adjusted to local values in the model simulations with the bespoke growing season (GS besp.). The results displayed in Fig. 8 allow for an assessment of over- or underestimation of ozone risk arising from the use of generic parameterizations established

from less climatically extreme conditions. Critical levels for reduction in biomass (deciduous forest $4\,\%$, coniferous forest $2\,\%$, and grasslands $10\,\%$ biomass reduction, Mills et al., 2017; Hayes et al., 2021) are shown as horizontal dashed lines. We identify that an increase in cold tolerance (MM $\rightarrow$ cold $\rightarrow$ subarctic) leads to higher ozone uptake in both probed years, while the usage of locally adapted GS causes a reduction. The earlier and longer GS in 2019 leads to increased ozone uptake compared to 2018 despite more episodes of elevated $[\text{O}_3]$ in 2018. Due to the shape of $f_{\text{light}}$, a symmetric variation leads to





an asymmetric response in $POD_1$. We find that an opening of stomata at lower light intensities can cause higher sensitivity to drought conditions. Similarly, a temperature acclimation to cold or subarctic conditions can lead to an amplification of effects by droughts.

The magnitude of these effects varies between PFTs as well as years, but the predicted ozone uptake for the bespoke temperature parameterization is always larger than for the MM parameterizations and of the same order of magnitude as the

variability between the years studied here. Our bespoke temperature parameterizations suggest considerable underestimations of ozone risk for subarctic species when relying on the generic MM parameterizations.

For deciduous trees (Fig. 8a) all $POD_1$ simulations exceed the CL by $5-25\,\mathrm{mmol\,O_3\,m^{-2}}$. Deciduous trees display the largest variation of ozone uptake among all PFTs ($10-31\,\mathrm{mmol\,O_3\,m^{-2}}$). In particular, the sensitivity to the light parameterization is pronounced. With MM GS, $POD_1$ values were higher in 2018 than in 2019, throughout our analyses. However,

taking the bespoke GS into account, this is reversed for the subarctic temperature acclimation. This can be explained by more favorable growing conditions in 2019. Low soil water potential reduced the $POD_1$ values in 2018 in some cases when using the MM growing season, but did not or only slightly when using bespoke GS. The temperature acclimation-dependent difference in ozone uptake deduced for bespoke GS ranges between $(1.4-3.6)\,\mathrm{mmol\,O_3\,m^{-2}}$ in 2018 and $(2.7-7.7)\,\mathrm{mmol\,O_3\,m^{-2}}$ in 2019. The maximum deviations deduced from the $-20\,\%$ (superscript) and $+20\,\%$ (subscript) variation of $f_{\mathrm{light}}$ amount to

$^{+1.6}_{-2.3}\,\mathrm{mmol\,O_3\,m^{-2}}$ for 2018 and $^{+1.9}_{-2.7}\,\mathrm{mmol\,O_3\,m^{-2}}$ for 2019.

For coniferous trees (Fig. 8b), most simulations slightly exceed the CL ($0-10\,\mathrm{mmol\,O_3\,m^{-2}}$). The MM Boreal Norway spruce parameterization was already well acclimated to the prevailing temperatures and therefore differs the least from our bespoke parameterizations. Taking the bespoke GS and canopy height into account, we find a higher ozone uptake in 2019 than in 2018. The overall difference deduced from bespoke GS for subarctic and cold parameterizations with respect to

MM range between $(0.5-1.0)\,\mathrm{mmol\,O_3\,m^{-2}}$ in 2018 and $(2.0-3.3)\,\mathrm{mmol\,O_3\,m^{-2}}$ in 2019. The maximum deviations deduced from the $-20\,\%$ (superscript) and $+20\,\%$ (subscript) variation of $f_{\mathrm{light}}$ and amounts to $^{+1.0}_{-0.8}\,\mathrm{mmol\,O_3\,m^{-2}}$ for 2018 and $^{+1.3}_{-1.0}\,\mathrm{mmol\,O_3\,m^{-2}}$ for 2019.

For perennial grassland (Fig. 8c), all simulations with bespoke GS stay below the CL. Perennial grassland shows the smallest ozone uptake, but a similarly large response to the temperature parameterizations as deciduous trees. Again, we find a reversal

for predicted ozone damage risk in $POD_1$ between 2018 and 2019 for the subarctic type. Perennial grassland shows the lowest sensitivity to the light threshold and SWP is only relevant without bespoke GS. The overall difference deduced from bespoke GS for subarctic and cold types with respect to MM ranges between $(2.2-3.4)\,\mathrm{mmol\,O_3\,m^{-2}}$ in 2018 and $(4.4-6.4)\,\mathrm{mmol\,O_3\,m^{-2}}$ in 2019. The lower maximum temperature tolerance in the subarctic compared to the cold temperature parameterization probably caused the lower $POD_1$ values calculated for 2018 in subarctic compared to cold climate, whereas

this was not seen for 2019. The maximum deviations deduced from the $-20\,\%$ (superscript) and $+20\,\%$ (subscript) variation of $f_{\mathrm{light}}$ amount to $^{+0.3}_{-0.2}\,\mathrm{mmol\,O_3\,m^{-2}}$ and $^{+0.3}_{-0.3}\,\mathrm{mmol\,O_3\,m^{-2}}$, respectively.

From the difference between 2018 and 2019, we infer a substantial interannual variability. The maximum deviation from the generic default parameterizations due to our bespoke temperature and light response functions is of the same order of magnitude as the interannual variability.





**Table 4.** Estimated total biomass reduction in % for temperature acclimation *subarctic* with bespoke GS and SWP=off. The uncertainty ranges reported here correspond to maximum a divergence deducted from varying $f_{light}$ and SWP=on. For comparison, the corresponding biomass reduction for the generic default MM averaged over both years is shown. The reported standard deviation is computed from all sensitivity simulations ($\pm 20\%$ in extent of stomatal opening at low light, SWP on/off, and bespoke GS).

| Year | PFT | | | |
|---|---|---|---|---|
| | Deciduous tree | Coniferous tree | Perennial grassland | |
| | | | total | above ground |
| 2018 | $15.5^{+(1.9...2.1)}_{-(0.8...1.5)}$ | $2.5^{+0.2}_{-0.2}$ | $9.7^{+0.2}_{-0.2}$ | $12.4^{+0.2}_{-0.2}$ |
| 2019 | $17.4^{+2.5}_{-1.8}$ | $3.0^{+0.2}_{-0.3}$ | $10.5^{+(0.1...0.2)}_{-(0.0...0.1)}$ | $13.7^{+(0.1...0.3)}_{-(0.1...0.3)}$ |
| $\langle MM \rangle$ | $11.2 \pm 1.1$ | $2.31 \pm 0.04$ | $7.5 \pm 0.9$ | $9.6 \pm 1.4$ |

We estimated the biomass reductions in accordance to the CLs (Mills et al., 2017; Hayes et al., 2021) for the most extreme temperature parameterization (*subarctic*) for each bespoke PFT. The estimates are listed in Table 4. The reported range corresponds to the maximum deviation defined by the $\pm 20\%$ variation of $f_{light}$ with SWP=on. The total biomass of deciduous trees and perennial grassland was substantially reduced in comparison with the default MM parameterizations in both years. The CL for perennial grassland is higher than for the other species. From an agro-economical point of view in our focus area, where the grass is cut only once a year, a loss of biomass of more than $10\%$ can have severer consequences compared to more productive areas in central Europe and may need adjustment.

Coniferous trees are the least affected and the magnitude of total biomass reduction is almost independent of the choice of the parameterization within given uncertainties. This may reflect that the temperature parameterization in the MM is rarely a limiting factor for stomatal opening.

## 5 Discussion and conclusions

To summarize, we have developed bespoke parameterizations of common vegetation types (coniferous and deciduous trees, perennial grassland) to a subarctic climate and studied their effect on $POD_y$. The comparison between meteorological conditions in 2018 and 2019 and their divergence from climatology allowed us to assess the influence of key environmental variables such as temperature, PPFD, and precipitation on vegetation susceptibility to $O_3$ damage in light of future changes as may occur under climate change. We found that conditions for ozone formation were more favorable in the 2018 growing season than in 2019, with 2018 being significantly warmer and less cloudy in spring and early summer. Accordingly, peak ozone concentrations occurred more frequently and at higher levels in 2018. This was particularly a result of the extended heatwave in spring and early summer and associated, extensive wildfires in Central Sweden. However, ozone concentrations during the growing season were not significantly enhanced and $POD_1$ values were very similar for both years. This suggests that the overall length





of the growing season is more crucial in risk assessment when using flux-based metrics than episodes of peak concentrations as these peak [$O_3$] often coincide with environmental conditions that limit ozone uptake in the $DO_3SE$ model.

Based on a MODIS photosynthesis product (Running et al., 2015), we estimated the growing season for coniferous trees in 2018 to be at least one month shorter than in 2019. The determined start of growing season in 2018 (doy 122) and 2019 (doy 106) lies within the range given by observations in Fennoscandia (Kolari et al., 2007; Karlsson et al., 2018). We deduced

the start of the growing season for deciduous trees based on gridded observational temperature and for perennial grassland on snow depth data. For perennial grassland, we assumed an one-month latency period after snowmelt that is supported by observational data from Rovaniemi (Finland) (Korhonen et al., 2018). With respect to ongoing climate change, a clear positive trend emerged in length ($5.2\,\mathrm{d\,decade^{-1}}$) of the growing season that is almost equally distributed between earlier start ($2.9\,\mathrm{days\,decade^{-1}}$) and later end ($2.3\,\mathrm{d\,decade^{-1}}$) (Appendix Fig. A1).

The $DO_3SE$ modeling results are in contrast to our observation of more visible damage in 2018 compared to 2019. However, visible damage caused by peak ozone uptake and dependence of ozone sensibility on phenology are not accounted for in $POD_y$. In this regard, open-top chamber (OTC) experiments performed in northern Finland where Scots pine and downy birch seedlings were exposed to elevated ozone concentrations attributed a reduction in biomass and reproduction with visible damage explicitly to peak $O_3$ concentrations and fast phenological development at high growth rate (Manninen et al., 2009). In

particular, forbs and perennial grassland are more susceptible to ozone-induced damage in the reproductive state (Bassin et al., 2004). That might explain why we observed damage in the ozone garden to a larger extent in 2018 than in 2019. An associated damage function to ozone depending on leaf age as proposed by Musselman et al. (2006) or phenological state might improve the predictions.

At the same time, intra-species variability even at the same location is typically non-negligible (Bassin et al., 2004) but not

accounted for in the models. The literature results regarding the ozone sensitivity of natural vegetation in northern Europe are contradictive. Subramanian et al. (2014) reported a higher, modeled reduction of net primary production in conifers ($4.3 - 15.5\,\%$) than birch ($1.4 - 4.3\,\%$) under elevated ozone in Sweden, while Girgždienė et al. (2009) observed more visible damage on deciduous trees than on Norway spruce in Lithuania. But visible damage and biomass reduction may not occur at the same rate and hence may not be interchangeable.

In our study, we do not explicitly account for this natural distribution of individual plant traits, but our bespoke response functions reflect the diversity within a given species. In particular, we assumed two tolerance levels to cold temperatures than for central European species. To this end, we computed a PDF from observed temperature at Svanhovd (May–September 1992–2000). We adjusted the parameters in the temperature response function of stomatal conductance to increase the enclosed area with the PDF. Assuming a proportionality between $g_{sto}$ and $A_n$, this is equivalent to optimizing carbon uptake. Similarly, we

deduced a PPFD PDF and varied the light response function to test a higher or lower extent of stomatal opening at low light intensities. Remark that these parameterizations are hypothetical and need verification by experimental data. We found that soil water potential under 2018/19 meteorological conditions was negligible when considering the bespoke growing season. We have not assessed the effect of the plants' response to VPD in detail, but our climatological data indicated that conditions at Svanhovd are shifting towards a more VPD limited regime.





Our key findings are:

1. Our bespoke parameterizations for subarctic species are better suited to describe vegetation adapted to subarctic conditions for instance allowing for stomata opening during a larger part of the growing season,

2. the parameterizations defined in the mapping manual for European regional risk assessment appear to not adequately capture the physiological responses of subarctic vegetation that are important determinants of $[O_3]$ susceptibility and are likely to underestimate ozone-damage risk, and

3. climatic conditions that are promoting a longer growing season are likely to increase the ozone-damage risk in flux-based metrics.

In this regard, vegetation might become more exposed to higher ozone concentrations in early phenological stages due to an overlap with ozone spring peak (Karlsson et al., 2007). However, the decline of this ozone spring peak is partly caused by the uptake of vegetation. It, therefore, remains unclear whether an earlier start of the growing season will increase the exposure of vegetation to ozone or lead to compensation due to an earlier decline of the peak ozone in the future. Hayes et al. (2021) pointed out that the highest susceptibility of vegetation to future ozone variability is likely to occur in summer when vegetation is most productive. The actual resilience of northern Fennoscandia plant communities to climate change strongly depends on both the range of existing acclimations as well as the actual type of acclimation. This means vegetation that is acclimated to cold temperatures and little to heat- and drought stress might suffer more strongly from a more frequent occurence of heatwaves than plants that have a higher temperature tolerance.

Following the methods described in Mills et al. (2017), we estimated biomass reductions due to ozone uptake for both 2018 and 2019 and found substantial reductions for deciduous trees ($15.5 - 17.4\%$) and perennial grassland (above ground, $12.4 - 13.7\%$) in both years. These reductions are higher than for the default parameterizations, $11.2\%$ (deciduous) and $9.6\%$ (perennial grassland). For coniferous trees, the estimated reduction in biomass differs insignificantly between our bespoke parameterizations ($2.5 - 3\%$) and the default ($2.31\%$). This indicates a higher modeled robustness to ozone damage in coniferous trees. All these results have to be treated with care because the biomass reduction functions were established based on studies from less extreme climates. In particular, CLs for deciduous and coniferous trees were breached in all years and for all parameterizations. For perennial grassland, the CL was not breached if a bespoke GS was assumed, but the mapping manual defined CL of $10\%$ loss in biomass might not be transferable to subarctic regions with considerable constraints on productivity leading to more severe economical consequences.

Beyond the risk assessment of ozone-induced damage, our bespoke temperature response functions, once verified by observations, can have important implications on land-surface modeling in global models, where problems in productivity of species especially in the Arctic regions occur due to PFTs which are not suitable for the climate. We pointed out, concerning default perennial grassland parameterization, that these perennial grasslands did not show any substantial stomatal conductance in 2019 characterized as a normal year. In terms of GPP this would mean unrealistically low carbon uptake. Automation of the here proposed PDF-based acclimation using machine learning techniques could overcome these issues in the future.





*Data availability.* All data is available from public databases or through institutional access. DO$_3$SE modeling results can be made available upon request.

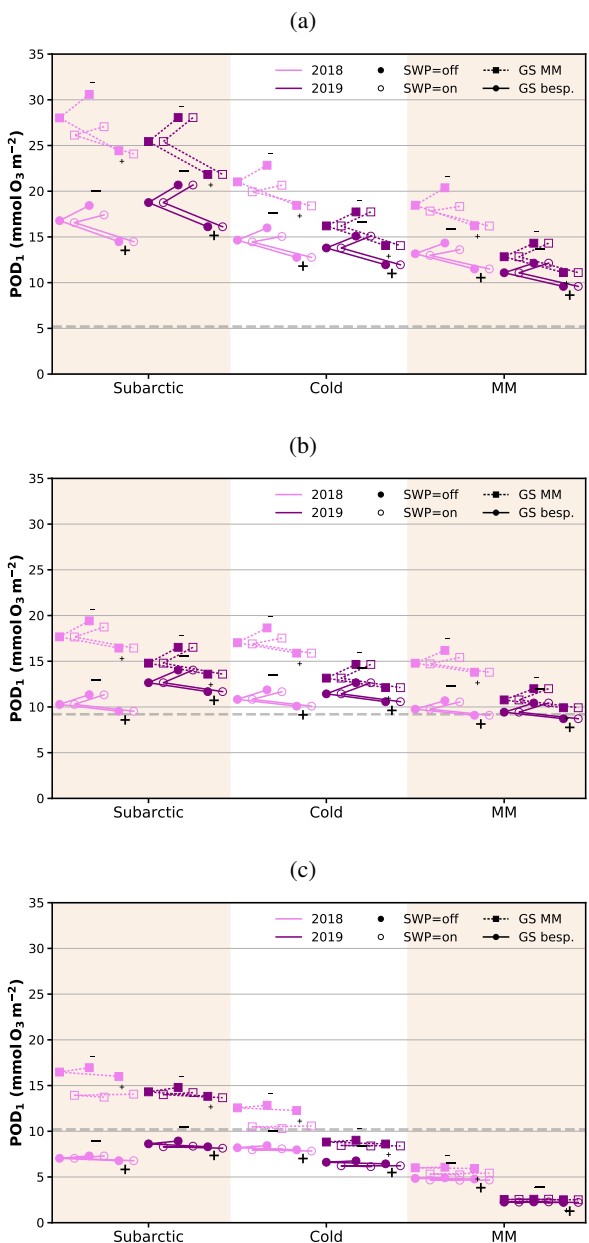

**Figure 8.** DO$_3$SE modeling POD$_1$ results for (a) deciduous trees, (b) coniferous trees, and (c) perennial grasslands at Svanhovd in 2018 (pink) and 2019 (purple). The results varied depending on parameterizations of temperature response functions (MM, Cold, and Subarctic, details in Table 1), growing season (GS MM (squares) or GS bespoke (circles), details in Table 3), light response functions (first, second and third point attached by a line are depicting the MM, PPFD0.8 and PPFD1.2, details in Table 2), and the effect of taking SWP into account (filled symbols without and open symbols with effects of drought kept in the model). Critical levels for ozone damage (Mills et al., 2017; Hayes et al., 2021) are given as dashed horizontal lines. Note that the horizontal axis is only categorical.





**Appendix A: Growing season**

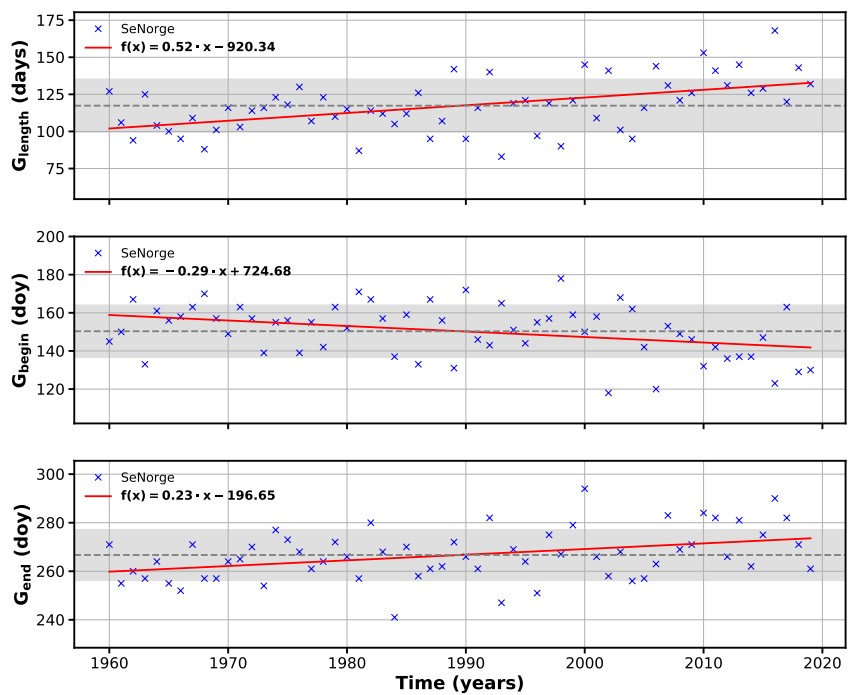

**Figure A1.** Estimated shift and prolongation of growing season at Svanhovd over the past 6 decades based on data from NVE, Meteorologisk Institutt, Kartverket (2020).

**Appendix B: DO3SE model**

**B1 Model description**

The opening of stomata is governed by different species-specific and environmental factors and can be described by a multiplicative model (Jarvis, 1976; Emberson et al., 2000; Mills et al., 2017):

$$g_{sto}^k = g_{max}^k \cdot f_{phen}^k \cdot f_{light}^k \cdot \max\left\{ f_{min}^k, f_T^k \cdot f_{VPD}^k \cdot f_{SWP}^k \right\}. \tag{B1}$$

Where $g_{max}$ is the specific-specific maximum stomatal conductance value that is then modified by seasonal and environmental factors that vary within a range $0 - 1$. They are empirically determined and account for leaf phenology ($f_{phen}$), light ($f_{light}$), temperature ($f_T$), water vapor pressure deficit ($f_{VPD}$), and soil water potential ($f_{SWP}$). All factors differ with plant functional type (PFT) denoted with $k$.





The temperature adjustment function is defined as

$$f_{\mathrm{T}} = \frac{T_{\mathrm{air}} - T_{\min}}{T_{\mathrm{opt}} - T_{\min}} \cdot \left( \frac{T_{\max} - T_{\mathrm{air}}}{T_{\max} - T_{\mathrm{opt}}} \right)^{\beta}, \tag{B2}$$

with $\beta = \frac{T_{\max} - T_{\mathrm{opt}}}{T_{\mathrm{opt}} - T_{\min}}$ a mixed reciprocal polynomial function of temperature. The shape parameters $T_{\min}$, $T_{\max}$ and $T_{\mathrm{opt}}$ are tabu-
lated for various PFT and $T_{\mathrm{air}}$ is the near-surface air temperature. All temperatures are defined in units of $^{\circ}C$.

    The water vapor pressure deficit function is

$$f_{\mathrm{VPD}} = f_{\min} + (1 - f_{\min}) \cdot \frac{D_{\min} - \mathrm{VPD}}{D_{\min} - D_{\max}} \tag{B3}$$

where VPD is the leaf to air vapor pressure deficit in kPa with $f_{\min}$, $D_{\min}$, $D_{\max}$ describing the relative stomatal conductance
to changes in vapor pressure deficit.

    The wavelength band $400 - 700\,\mathrm{nm}$ plant chlorophyll responds to is called photosynthetic active radiation (PAR). Its integral
is the photosynthetic photon flux density (PPFD). The relationship between relative $g_{\mathrm{sto}}$ and PPFD is given by

$$f_{\mathrm{light}} = 1 - \exp(-\alpha_{\mathrm{light}} \cdot \mathrm{PPFD}) \tag{B4}$$

where $\alpha_{\mathrm{light}}$ is a slope parameter describing the extent of stomatal opening at low light intensities.

    The DO$_3$SE model as described in Büker et al. (2012) is used to simulate SWP across a PFT specific root depth according
to the Penman–Monteith energy balance method that drives water cycling through the soil–plant–atmosphere system. Büker
et al. (2012) discuss several formulations available to parameterize the reduction in stomata conductance due to water content
in the soil. Here we use the relationship between relative $g_{\mathrm{sto}}$ and soil water potential (SWP) given by:

$$f_{\mathrm{SWP}} = \min \left\{ 1, \max \left\{ f_{\min}, \frac{(1 - f_{\min}) \cdot (\mathrm{SWP}_{\min} - \mathrm{SWP})}{\mathrm{SWP}_{\min} - \mathrm{SWP}_{\max}} + f_{\min} \right\} \right\} \tag{B5}$$

where SWP is the soil water potential across the root zone and $\mathrm{SWP}_{\min}$ and $\mathrm{SWP}_{\max}$ are the parameters describing the $f_{\mathrm{SWP}}$
relationship.

    To compute POD$_y$, we estimate the stomatal O$_3$ flux ($\Phi_{\mathrm{sto}}$) based on the assumption that the concentration of O$_3$ at the top
of the canopy represents a reasonable estimate of the concentration at the upper surface of the laminar layer for a sunlit upper
canopy leaf. The terms $r_c$ (leaf surface resistance) and $r_b$ (quasi-laminar resistance) allow for the deposition of O$_3$ to the leaf
and the fraction that is taken up via the stomata.

$$\Phi_{\mathrm{sto}} = [\mathrm{O}_3] \cdot \frac{u(z_1) \cdot g_{\mathrm{sto}} \cdot r_c}{r_b + r_c}. \tag{B6}$$

The quasi-laminar boundary layer resistance is calculated by

$$r_b = 1.3 \cdot 150 \cdot \sqrt{\frac{L}{u(z_1)}}. \tag{B7}$$

Where $L$ is the cross-wind leaf dimension, $u(z_1)$ the wind speed at height $z_1$, and the factor $1.3$ accounts for the diffusivity
between heat and O$_3$.





## B2 Input data and gap filling methodology

The $DO_3SE$ model requires hourly, continuous meteorological observations. In addition to variables covered in Section 2, $2\,m$ wind $u_{2m}$ and vapor pressure deficit VPD are needed. VPD has been calculated from observed $T_{2m}$ and relative humidity

$$\text{VPD} = P_s(T) \cdot \left(1 - \frac{\text{relHum}}{100}\right), \tag{B8}$$

with saturation vapor pressure $P_s(T)$ in $hPa$ derived from Arden Buck equation (Buck, 1981; Buck Research Instruments, LLC, 2012).

The following gap filling methodology was devised for the meteorological input data:

– Single hours of missing data were filled by taking the average of the hourly values coming before, and after, the missing value.

– Several consecutive hours of missing data (23 or less) were filled by taking the average of the corresponding hour the day before, and the day after; and repeating this for each missing hour of data. If data were unavailable from that hour of the previous day, then only the value from the day after was used and vice versa. Interpolated values were not used in calculating averages.

– Data gaps longer than $24\,h$ were filled using weekly diurnal averages. i.e. an average was calculated using the corresponding hour throughout the week before and after. Interpolated values were not used in calculating averages.

For the missing ozone data, we applied a Reynolds decomposition of the form:

$$[O_3] = \langle [O_3] \rangle + \Delta[O_3], \tag{B9}$$

with climatology $\langle [O_3] \rangle$ and anomalies $\Delta[O_3]$ as described in Falk et al. (2021).

## B3 Initial $DO_3SE$ modeling results with mapping manual parameters

To demonstrate the necessary acclimation of the mapping manual parameters, we show the initial results of stomatal conductance at leaf-level ($G_{sto}^{leaf}$) and $POD_y$ over $doy$ for both years and all PFTs (Fig. B1). Observed $[O_3]$ is also indicated. From Fig. B1f) it is apparent that the mapping manual parameterized grassland would not have been able to grow in 2019.

## B4 Bespoke parameterizations

To assess the $G_{start}$ and $G_{end}$ of the growing season for coniferous trees at Svanhovd in 2018/19, we used the net photosynthesis product ($A_{net}$) of MODIS AQUA/TERRA over a $1 \times 1\,km$ area centered at Svanhovd. MODIS data indicate a higher photosynthetic activity in 2018 than in 2019. As shown in Fig. B2 we fitted a second order polynomial function of the general form

$$A_{net}(t) = -m_0 \cdot (t - m_1)^2 + m_2, \tag{B10}$$



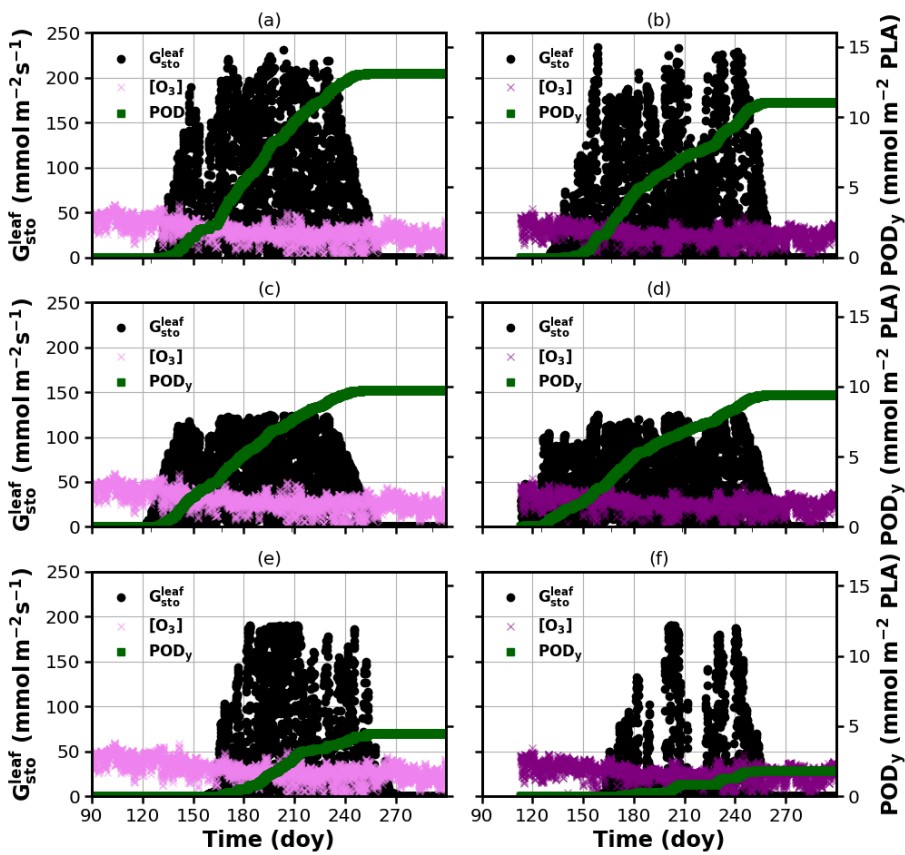

**Figure B1.** DO$_3$SE modeling results for mapping manual default parameterization. POD$_y$ is shown over doy, March–October. A flux threshold $y = 1\,\mathrm{nmol\,m^{-2}\,s^{-1}}$ per projected leaf area (PLA) has been chosen. [O$_3$] are plotted on the same axis and scales as $G_{\mathrm{sto}}^{\mathrm{leaf}}$ but in units of ppb. (a, b) deciduous tree; (c, d) coniferous tree; (e, f) perennial grassland. From left to right: 2018, 2019.

with the form parameters $m_i$ through the data. Numerically, we retrieved the roots as $G_{\mathrm{start}}$ 122/106 and $G_{\mathrm{end}}$ 261/274 for 2018 and 2019, respectively.

As described in Section 4.1, we developed bespoke parameterizations for natural and semi-natural vegetation at Svanhovd. Here, we show the temperature response and light response functions for coniferous and deciduous trees (Figs. B3-B4).





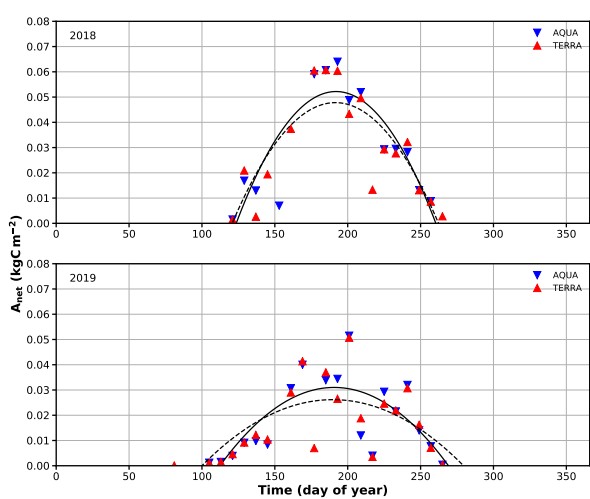

**Figure B2.** Estimated $G_{start}$ and $G_{end}$ of growing season for coniferous trees from MODIS Aqua/Terra net photosynthesis (PSN) product. A $1 \times 1\,\mathrm{km}$ area around Svanhovd was selected. Daily averaged data for both 2018 and 2019 has been fitted with a quadratic polynomial function. The numerically computed root yields: BGS doy 122/106 and EGS doy 261/274 for 2018/19, respectively.





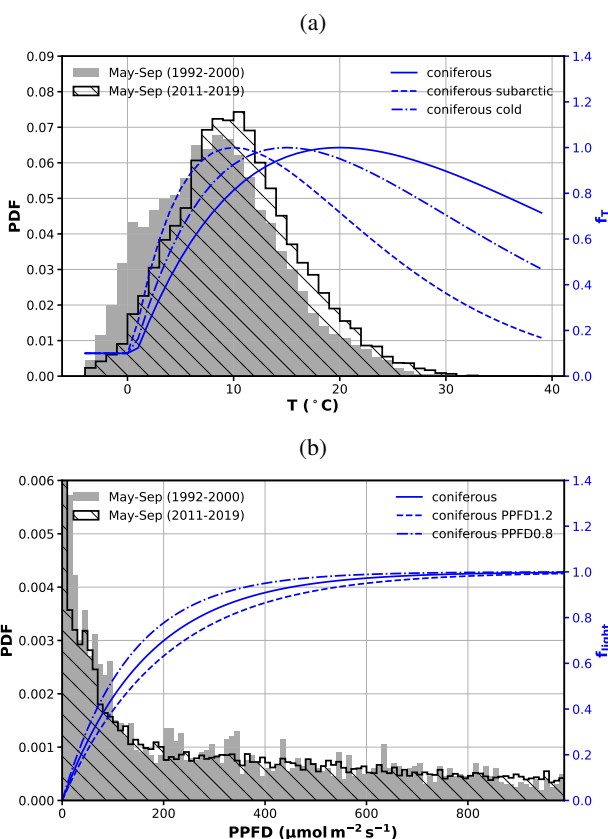

**Figure B3.** Construction of bespoke response functions for coniferous trees. (a) $f_T$ and (b) $f_{\text{flight}}$ are shown together with underlying $T_{\text{air}}$ and $Q_0$ climatologies (probability density function - PDF), respectively. Original mapping manual parameterization is shown in comparison as solid line. Note that $Q_0$ has been truncated to 0.006. PPFD0.8 and PPFD1.2 refer to $\alpha$ values increasing/decreasing PPFD at $f_{\text{light}} = 0.5$ by $\pm 20\,\%$, respectively.



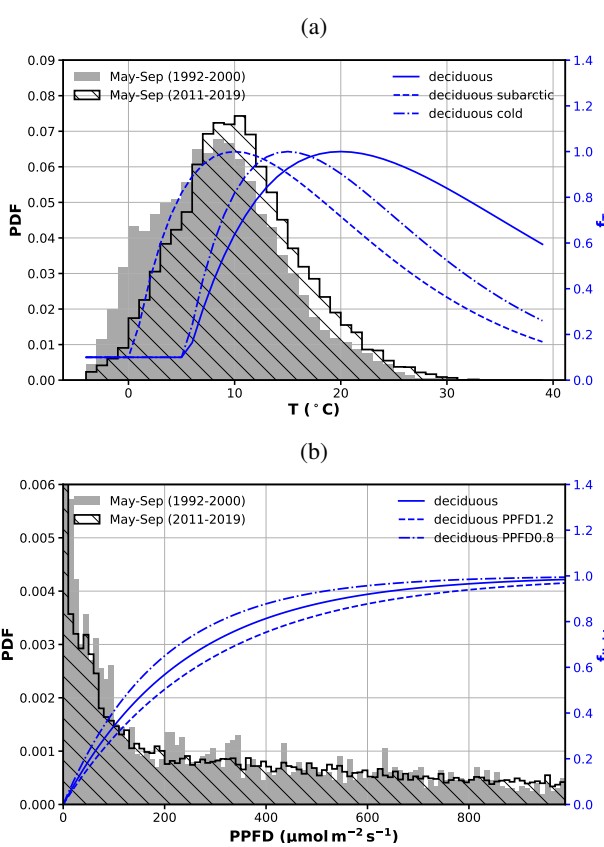

**Figure B4.** Construction of bespoke response functions for deciduous trees. (a) $f_T$ and (b) $f_{flight}$ are shown together with underlying $T_{air}$ and $Q_0$ climatologies (probability density function - PDF), respectively. Original mapping manual parameterization is shown in comparison as solid line. Note that $Q_0$ has been truncated to $0.006$. PPFD0.8 and PPFD1.2 refer to $\alpha$ values increasing/decreasing PPFD at $f_{light} = 0.5$ by $\pm 20\%$, respectively.



*Author contributions.* All authors contributed to conceptualization of this research article and commented on the manuscript. SF has prepared the original draft, collected and processed ozone and environmental data, and performed all statistical analyses. AVV has conducted the on-site observation of vegetation damage induced by ozone, provided advice on plant physiological processes, collected existing literature on subarctic vegetation, contributed significantly proofreading this research article. LE has contributed with expertise in $DO_3SE$ modeling and suggestion of the PDF-based temperature acclimation methodology. CO has collected PFT parameters from the literature, performed all $DO_3SE$ simulations and validation. AE contributed with her experience regarding subarctic vegetation in Finnmark. FS contributed with an assessment of the 2018 meteorological conditions. TB gave valuable guidance in a broader research sense. Funding acquisition for the project: AVV, AE, FS, TB.

*Competing interests.* The authors declare that they have no conflict of interest.

*Acknowledgements.* We thank NILU for performing the ozone measurements at Svanhovd for us, Bjørg Rognerud (Department of Geosciences, UiO) for processing SeNorge.no data with respect to the beginning of the growing season, Tore Flatlandsmo Berglen (NILU) for hourly pressure data from Svanhovd, NIBIO Environment Center Svanhovd for establishing and running the ozone garden both years, and Volkmar Timmermann (NIBIO) for sharing ICP Forest data from Svanhovd. This work was supported by the Research Council of Norway (Grant No. 268073).



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
