# Peer review of "Parameterization of the responses of subarctic European vegetation to key environmental variables for ozone risk assessment"

_Biogeosciences, 2021_

## Author Comment (AC1)

**Authors' response**

To bg-2021-260-RC2 (23 Nov 2021): We thank the anonymous referee #2 for their very constructive comments and useful suggestions. We will address all **general comments** raised regarding the structure and readability of the manuscript in our next revision. We will refine the language to reduce ambiguity and supplement our statements with appropriate citations where indicated by the referee.

As the manuscript will, hence, undergo major structural changes, we can not address all **specific / technical comments** in detail at this point. In the next revision of the manuscript, we will take all specific and technical comments which are still relevant into consideration.

In the following, we shall respond to all issues where reasonable.

**General comments**

- "[...] main figure is missing [...]" The referee is referring to Fig. 8 which is apparently absent from their version of the manuscript. We downloaded the pre-print from the associated discussion page (bg-2021-260) and cannot confirm Fig. 8 to be missing. It can be found on page 23 (just before the list of references). We, however, acknowledge that the current placement (after the main body of text) is rather unfortunate. This shall be fixed in the final typesetting process!

**Specific comments**

- "[...] I find the introduction to be a bit longer than would be most effective for conveying this information. It would be helpful to break it into subsections [...]. [...]content of [...] lines 115–134 is important for contextualizing [...] could be better emphasized [...] or [...] opening paragraph for [...] section 2 [...]." Thank you for this very constructive comment. We will condense our introduction and put subsections in place where necessary. As suggested, we'd move the discussion of meteorology to Section 2.

- "Line 102 – The concept of critical loads is much older than 2016 [...] please include an earlier citations [...]" Indeed the concept is much older (1970s). Critical load is a term defined by the UNECE's Convention on Long Range Transboundary Air Pollution (LRTAP) and is generally associated with wet deposition of N and S. For ozone (and other pollutant gases (i.e. not in solution) we refer to 'critical levels'. Either concept is the level or load of the pollutant below which damage or injury would not be expected to occur to the receptor in question (e.g. forests, crops, grasslands etc...). We will include an appropriate citation.

- "Please correct the text in line 105. [...] This doesn't seem to be a correct descriptor for Equation 2 [...]" Thank you for pointing out this carelessness which should not have gone unnoticed.

- "In the paragraph that starts at line 241, the meaning of the word "significant" is unclear and varies within the paragraph. [...] it should be stated early on in this section. [...] Ideally it will pertain to a significance test. [...] tie back what is said to the objective of the paper." Thank you. We will restructure the section to make the definition of the term "significance" easier assessable and unambiguous. The results shall be put into context with the paper's objective, either in-section or in a "results" Section. Regarding the significance test, we deliberately refrained from using p-values as their interpretation is often ill defined (for a discussion see statement of the American Statistical Association (2016)).

- "Section 4 - Please expand the description of the $DO_3SE$ model" A comprehensive model description can be found in Appendix B. As no modifications to the $DO_3SE$ model were made (except for the change in parameters), we had concluded that an excessive model description would distract the reader. In particular, relationships between plant physiological parameters and meteorological observations are described in Appendix B1. Model "high-level" input and its preparation is described in Appendix B2. We agree that important information is therefore missing from the main text. In the course of restructuring, we'd include a concise model description in Section 4.

- "$f_T$ hasn't been defined yet." "Is $f_T = f_{temp}$?" These two issues are related. Of course, $f_T$ is the same as $f_{temp}$ and we should not have used both at the same time. We remove this ambiguity.

- "Line 304 (and Line 237-328) Again "extreme" is ambiguous here and seems to differ in definition from previous use [...]." We will carefully evaluate the use of the term and define it appropriately.

- "Lines 306-307 – Please clarify which time frames are being compared related to what is meant by "subject to climate change". In other words, are you comparing 1990s to 2000s? Or 1990s + 2000s with implicit impacts of climate relative to the preindustrial era?" In the context of our study, the time frame for climate change is 2010s compared to early 1990s.

- "Line 317-318 – Mention that this is PPFD 0.8 and 1.2 in parentheses [...]" We add the suggested information to clarify our naming convention.

- "Line 321 appears to substitute for $f_{light}$" in Equation 4, not for as described in the text." Indeed this sentence is ambiguous, and should read: "First, we solve Eq. (3) for the MM default values of $\alpha_{MM}$ at $f_{light} = 0.5$" We should, perhaps, clarify that we use $\gamma$ synonymously with PPFD in this paragraph or substitute $\gamma$ with PPFD.

- "Line 331 – Is the low standard deviation specific to nighttime?" No, we have selected our sample (May–October), so that the vegetation will not experience "nighttime" between 5 am and 9 am. See, f.e. TimeAndDate.com/Svanvik.

- "I find the conclusion that the standard deviation indicating "higher robustness to variability in growing conditions" to sweeping relative to the evidence precented. [...]" This might be a misunderstanding due to inaccuracies in language? The robustness is not a conclusion but a presumption. We assume that the essential climate variables (excluding ozone) vary substantially in our sampled 20 years. Though, we have not explicitly shown the associated standard deviation of this particular subsample (5–9 am and 11 am–1 pm). We could do this to strengthen our point. A low standard deviation in stomatal conductance, therefore, suggests that stomatal conductance is relatively unaffected by these variations. We refer to this as "robustness". We will make this more clear in the next revision. Or remove the whole point as suggested.

- "Line 332 – "subarctic-PPFD0.8" is "best" relative to what?" In this case "best" refers to the highest relative stomatal conductance and smallest standard deviation compared to the other probed combinations of parameter. We rephrase the sentence accordingly.

- "Is there a citation or justification for the 1 nmol/m2/s flux threshold?" This threshold (amongst others) has been introduced for modeling ozone induced damage on forest as part of the ICP Vegetation taskforce in the framework of LRTAP. "A uniform $O_3$ flux threshold of $Y = 1$nmol m$^{-2}$ s$^{-1}$ PLA was adopted for use in POD$_Y$SPEC for all tree species at the $O_3$ Critical Levels workshop in Madrid, November 2016, based on data and analyses as presented in Büker et al. (2015). For the majority of tree species, this threshold fulfilled the recommendations that the confidence interval of the intercept includes 100% and that the $R_2$ value is within 2% of the maximum $R_2$ value (Büker et al., 2015)." (Mapping Manual; Mills et al. (2017)) We will include this in the revised manuscript.

- "Figure 8 appears to be missing from the document." As mentioned earlier, the placement of the figure on page 23 (after the main body of text) is unfortunate. The figure is not missing (at least as far as we can tell), but might, therefore, have slipped the vigilant eye of the referee.

- "Lines 367-372 – "leads to" seems strong, given that there are many variables changing [...]" We change the sentence accordingly.

- "Where is it shown that temperature acclimation relates to the amplification of drought effects?" This is shown as part of Fig. 8 in which we summarize all sensitivity studies. Drought affects the ozone dose in 2018 only and not in 2019. The effect is stronger the more acclimated to cold temperatures our parameterizations are.

**Technical comments**

We will appropriately address all comments raised, but only address selected items at this point.

- Concerning Figures 3–5 Thank you for the useful suggestions to further improve our figures. We will take them into consideration. We may kindly ask to review Figure 8 as well.

- I haven't previously seen this method for identifying the start and end points of photosynthesis, and (noting that I'm not an ecologist) it seems elegant in concept and approach. Thank you. This was a pragmatic, data-driven approach. To our knowledge, most satellite data based studies on start and end of growing season focus on correlations with essential climate variables to identify processes and relationships.

- "Sentence in lines 339–340: "This value will be used for all PFTs alike". Please specify – $A_{end}$? In this paragraph please explain why it is acceptable to use the same $A_{end}$ for all PFTs while $A_{start}$ is specific." It is still difficult to predict when deciduous vegetation will start the abscission process. For practical reasons, we set the date for all PFTs alike, although it could differ in reality. Coniferous trees are capable of photosynthesis as long as there is light which is the hard limit for all species alike in our focus area. Grassland, on the other hand, will in most cases stop photosynthesis after it has been covered by snow.

- "[...] please clarify what leaf length you are using [...]" We have conducted two sets of simulations, one with leaf dimensions, tree height, and growing season as defined in the mapping manual and one with leaf dimensions of deciduous trees and hight of trees adjusted according to observations. We have not adjusted the leaf dimensions of coniferous trees and grassland. We will make this clearer.

- "Line 457 – do you mean the response to SWP, rather SWP itself being negligible?" Of course! We apologize for the sloppy language.

---

## Author Comment (AC2)

**Authors' response**

To bg-2021-260-RC1 (05 Nov 2021): We thank the anonymous referee #1 for their comments. We will address all **general comments** raised regarding the structure and readability of the manuscript in our next revision or resubmission. We consider separating the manuscript in Introduction–Methods–Results–Conclusions and will improve on the "methods" Section to make our results more comprehensible. We will refine the language to reduce ambiguity and supplement our statements with appropriate citations that were indicated by the referee.

As the manuscript will, hence, undergo major structural changes, we will not address all **specific / technical comments** in detail at this point. We will take all specific and technical comments which are still relevant into consideration.

In the following, we shall give a brief response to all relevant issues.

**General comments**

- "the [...] study is highly speculative and not based on any experimental evidence [...]" In the revision of our work, we will emphasize that the main objective of our work is to explore the possibilities of readily accessible data (i.e. long-term meteorological temperature observations, $CO_2$ flux estimations from remote sensing) to provide alternatives to site-specific data that is needed to parameterize plant physiology in models such as the $DO_3SE$ model used in our study.

- "Many claims sound superficial [...], such as [...] the alleged deviation of the years 2018 and 2019 from the site's climatology [...]" We acknowledge that we have not used tools such as the *p-value* (for a discussion see statement of the American Statistical Association (2016)). We will make our statistical approach clearer where necessary (see also more detailed responses below) and replace terms such as "significantly different" where significance has not been shown appropriately.

- "The manuscript is too long and the organization is confused [...] no clear division between methodological section and results section" We will address this in the next revision by fracturing the manuscript clearly into separate sections containing methods, results, and discussion.

**Specific comments**

- Lines 51-52. "and leads to a build-up of ozone and its precursors during winter." Are you sure? How can BVOC accumulate in the atmosphere if the vegetation is covered by snow? Please add some citations to support this claim. Much more credible is the following explanation based on stratospheric intrusions. We agree with the referee that vegetation that is covered by snow would not release BVOCs into the atmosphere. We implicitly assume this. Therefore, we did not explicitly state that BVOCs are not among the ozone precursors that accumulate in winter. We will rephrase the sentence accordingly: "[...] ozone and its non-biogenic

precursors [...]"

- Line 63. "the time in which vegetation can accumulate ozone." It sounds very bad written this way: vegetation does not accumulate ozone because ozone it is not bioaccumulative. Did you mean the dose? Thank you for pointing this out. We will rephrase accordingly and refer instead to the accumulated ozone dose.

- Line 77. "($\langle[O_3]\rangle$ = 36-54ppb)" Please explain the formalism. What do different brackets mean? We will make this notation and formalism clearer in the text. The "$[X]$" notation commonly refers to the concentration of a chemical species X. Parts-per-billion (ppb) is a volume mixing ratio (VMR) and strictly speaking no concentration. Though, if you calculate the amount of ozone per volume from ppb to $\mu g\,m^{-3}$ meter, assuming a pressure of 1 atmosphere, the temperature of 298K and use the ideal gas law, you get a factor $\sim 2$. Therefore, concentrations and VMRs are often used synonymously.
  We could also use another terminology, e.g. $\chi_{O_3}$ (e-education.psu.edu).
  The "$\langle A\rangle$" notation is derived from Dirac's "bra-ket" formalism in quantum mechanics, e.g. $\langle\Psi|A|\Psi\rangle \to \langle A\rangle$ "this expression gives the expectation value, or mean or average value, of the observable represented by operator A for the physical system in the state $|\Psi\rangle$ " (wikipedia).

- Line 81. "A substantial body of evidence exists that suggests flux-based metrics, that relate stomatal ozone uptake to vegetation damage, are biologically more relevant for risk assessments than exposure-based metrics." Well, please cite at least some works of this "substantial body" Thank you, we will include relevant citations.

- Figure 1. This figure was never referred in the text. This Figure has been referred to in Sect. 2 L177/179.

- Figure 2. It does not seem to me that the O3 concentrations of 2019 are different from those of 2018. The spring peak could even be identical (although unknown, because in 2019 O3 measurements started about 20 days after the spring peak) In Sect. 2, we assess in detail whether and how those two years differ. We will make this analysis more comprehensible and comprehensive in the next revision.
  Regarding the spring peak, until the beginning of May most of the vegetation in the subarctic is still either covered by snow (see Table 3 for the dates) or bud burst has not yet occurred. Hence, it was unlikely for deciduous vegetation and scrubs or other short vegetation to experience the ozone spring peak in 2018/19. We will come back to coniferous trees later.

- Line 146. Please make clear the acronym PFT on first use Plant functional type (PFT) was introduced in the abstract. Following the common code of conduct, we ought to introduce it also in the main matter. Thank you for pointing this out.

- Line 165. "luftkvalitet.no" What is it? And EBAS? Please make them clear. "luftkvalitet.no" is the web interface to the database of air quality measurements

in Norway operated by NILU. EBAS is a database with a web interface operated by NILU as well. It comprises global air quality measurements, e.g. trace gases, pollutants, particular matter from different networks. We will clarify this.

- Line 182. "This indicates that the vegetation was more affected by ozone in 2018 than in 2019." Being affected by visible symptoms does not necessarily mean having suffered biomass or productivity reduction. It is correct that biomass reduction and visible damage due to ozone are not the same. We discuss this briefly in Sect. 5 but will elaborate on it to make the difference clearer.

- Line 190. "high ozone concentrations ([O3] > 40ppb" It is strange to read that O3 concentrations above 40 ppb are "high" concentrations. Indeed 40 ppb is not particularly "high" compared to peak ozone pollution downwind of densely populated areas, e.g. in Eastern Asia. We will remove the term "high". However, ozone concentrations of 40 ppb have been chosen as a critical level for the metric accumulated over a threshold (AOT) to gauge ozone exposure toxic to ecosystems. This metric was first introduced at the UN/ECE-Workshop in Egham in 1992. For the protection of crops and semi-natural vegetation (limit to 5% yield reduction) and forest different targets exist (see e.g. EU directive).

- Line 194. "A method for gapfilling data has been presented in Falk et al. (2021)." Ok, but was it then applied to this work? Please write it. We will rephrase: "A method for gap-filling data has been presented in Falk et al. (2021) and applied in this work."

- Line 205. "We evaluate the statistical significance of divergences from the norm in these variables (referred to as anomalies) in 2018/19" I suspect a misuse of the locution "statistical significance". How was this significance assessed? Which statistical test was applied? What is the level of significance? We will elaborate on the hypothesis test in the next revision and may show the residuals as violin plots to increase the transparency of our method and make the data more assessable to the reader.
  In the recent version of the manuscript, we computed the excess number of days with residuals larger than $1\sigma$ first. This means we calculate the daily mean and std for 21 years $\times$ 24 hours (264 data points). From this, we find the residuals in 2018/19. In the next step, we explicitly assume that these are normally distributed – we will elaborate on this assumption in the next revision and check the underlying distributions (but see also Fig. 4). Due to natural variability (diurnal, annual) in the data, we chose a $1\sigma$ level for the significance test. Because we are not looking at repeated experiments with well constraint conditions, a much higher sigma level is not necessarily meaning full. But the diurnal cycle could be perhaps removed to smooth the data. Second, by grouping the data into months, we find the number of days with observations sufficiently diverging from the mean in each month. We count the number of days with residuals exceeding $1\sigma$ and compute the ratio with the number of days of observation. If this data were normally distributed, we

would expect 18% of data in this interval. If the percentage of days exceeding $1\sigma$ is larger, we state that we find a prominent excess of days with high ozone.

- Lines 206-208. I do not understand. Please, explicit the methodology. We will rephrase and include a brief description of the method. We use a Reynolds decomposition on the 1986-1996 data from Svanvik and data from all long-term monitoring stations in Fennoscandia (1992-2012, Jergul-Karasjok, Pallas, Esrange) to establish multi-annual means. We then remove the annual cycle from the July 2018 observations at Pallas (reference station) to find the anomalies. We have shown that these correlate best with observations at Svanvik. We find the time lag between the observations and shift the time series of anomalies accordingly. We then add the climatology of Svanvik to retrieve a reconstruction of the missing data.
  As 2019 does not deviate much from the norm by means of temperature, precipitation, and irradiance, we assume it for a normal year. From this, we find a probable offset of the historical ozone climatology compared to the present day and correct for this. We compared this reconstruction with high-resolution air quality model reanalysis and found a similar accuracy compared to observation. For details, please refer to the cited work Falk et al (2021).

- Line 213. "Averaged monthly accumulated precipitation (blue bars) is shown with standard deviation" It is not consistent to show SE once and STDEV the other time. The use of SE is more appropriate when estimating averages. We will change the figure accordingly.

- Line 228. "Darker colors indicate higher probability to observe these values."
  Line 229. "On top of the density distributions, a 10 days average of daily mean (h[O3]i10d) is displayed together with 1sigma uncertainties and SE, respectively" What does it mean? It is not clear to me. Why show a probability density if you are plotting a multiannual average? Or does the line represent the median instead? In Fig. 4, we show the distribution of ozone data in 2-dimensions. The dotes represent the daily averages (but only every 10th day is shown). We will rephrase the caption to correct this. We show the daily average to infer that not all daily mean ozone is strictly normal distributed (tails in one or the other direction) but it is also visually deductible that most data fall in the $1\sigma$ band justifying the chosen confidence intervals for a normal distribution. We shall, however, elaborate on this by statistical means.

- Line 232. "The decline in h[O3]i coincides with the average beginning of CO2 uptake by coniferous trees (Kolari et al., 2007; Wallin et al., 2013)" I didn't know that evergreens only uptake CO2 starting in May. I was convinced they always did. Is it true? Doesn't that contradict what you wrote in line 309 ("We base our temperature acclimation of coniferous trees on experimental results on Norway spruce which were found to be active already at rather low air temperatures and can reach 60% photosynthetic activity as early as doy 100 (Kolariet al., 2007;

Wallin et al., 2013).")? Here you state that photosynthesis is already active at DOY 100 and is at 60% of its maximum! Coniferous trees can indeed, in principle, take up $CO_2$ also in winter and the fact that they can suffer from something called frost drought shows that. Concerning the spring peak that was mentioned earlier, this means that coniferous trees may experience these elevated ozone concentrations regularly. Concerning the contradiction in the statements – if the conditions are right, coniferous trees can quickly reach 60% of their max photosynthesis but stop photosynthesis the days after. Conditions to keep up these levels of photosynthesis, however, would only occur from doy 100 onwards. To make sure that we do not falsely accumulate too much ozone from the spring peak period (when coniferous trees show on-off photosynthesis), we consulted satellite observations to extrapolate the dates when vegetations' photosynthesis (including coniferous trees) at Svanhovd rose over the detectable threshold.

- Line 233. "In July–September (doy 182–273), ozone is occasionally almost completely depleted. This hints to ozone uptake by vegetation even at low light intensities during midnight sun conditions in combination with stable planetary boundary layer conditions preventing mixing of ozone rich air." I don't understand the connection. What does the night uptake have to do with the occasionally complete ozone depletion? This was a working hypothesis and should be removed from the text.

- Line 244. "if a normal distribution is assumed" Are you sure that the distribution is normal and not lognormal or something else? There are some literature on the type of statistical distributions for variable such as Temperature, Rain, etc ... Moreover, looking at your Figure 6b the distribution of the irradiance seems to be a Poisson distribution. The data shown in Fig.6 are not the same as used to calculate the results for Fig.5 because the data selection is different. But we should of course check and describe the underlying distribution of the data selection used in Fig.5.

- Figure 5, caption. "dashed lines indicates statistical significance" Statistical significance of what? By means of what test was it obtained, at what alpha level? And what are the numbers on the top right of each graph? Regarding the test, see the response to "L205" above. The number at the top right is described in the caption: "The annual positive/negative deviations are indicated in the respective corners (right upper/lower)." This is the percentage of days on an annual basis that exceed the $1\sigma$ level. This indicates the overall deviation from a normal year. Only temperatures in 2018 were abnormally high compared to the climatology on the $1\sigma$ level.

- Line 251. "deviated significantly from the climatology on the 1 sigma level." Here the standard deviation is used as reference for the significance. But the significance of the deviation should be statistically tested in another way. Regarding the test, see the response to "L205" above. Our hypothesis test is that you would expect

$\approx 18\%$ of days exceeding the $1\sigma$ level (normal distribution). If we observe a higher percentage of days, we find an exceedance from the expectation. If this is not proper in terms of statistical terminology we can remove the term "significant" and call it exceedance (in case this is deemed a better term).

- Line 262. "We use the bias-corrected and cross-calibrated ozone climatology (Falk et al., 2021) and assess the monthly significance of the ozone concentration anomalies in 2018/19." "Bias-corrected cross-calibrated" ozone? What is it? And what is the "significance" of the concentration anomalies? Please explain. Please see the response to "Lines 206-208.". Cross-calibrated means that we used data from other monitoring stations in the region to reconstruct the missing data at Svanvik. Regarding "significance", see the explanation above.

- Line 267. "Further, we presume that fVPD and fSWP suit our vegetation types and no acclimation is necessary for these." This statement is questionable, because in cold conditions VPD can be high (you also told it in the conclusions) and the water in the soil can be limiting because partially unavailable due to freezing or other. The referee is correct that VPD can be high due to cold conditions and thus become a limiting factor. We did not find any unexpected or abnormal behavior of stomatal conductance concerning VPD. This indicates that the parameterized VPD limitation on stomatal conductance might be applicable for high VPD caused by either drought or cold. The response of subarctic species to high VPD or SWP could, however, differ. We have to assume that these parameterizations are fair enough. Different thresholds for $f_{\mathrm{VPD}}$ could have been tested in analogon to $f_{\mathrm{light}}$. Water (un)availability due to frozen soil is currently not represented in the $DO_3SE$ model and would be worthwhile investigating further.

- Line 272. "but a substantially higher number of peak [O3] were observed in 2018 than in 2019." How can you tell it if O3 measurements for all months of March, April and July are missing in 2019? I don't seem to see any differences between 2018 and 2019 As we have indicated, the measurement downtime in March and April was planned because most vegetation (in particular the one in the ozone garden) was covered by snow and hence not photosynthetically active (as the referee remarked in their comment regarding BVOC emission). "but a substantially higher number of peak [O3] were observed in 2018 than in 2019." refers back to Sect. 2 L188-195. Peak concentrations were calculated as exceedance over 40 ppb in summer, explicitly excluding the spring peak. Even with two weeks of data missing July 2018, we found 50 elevated ozone events in 2018 compared to 18 times in 2019. This should qualify as a "substantially higher number". This explanation will be added to the text for clarification.

- Line 291 " Note, however, that these parameterizations are hypothetical and have yet to be verified by experiments." Figure 6a. Looking at the graph I understand that you assume an adaptation of the subarctic grasslands to the temperature distribution of the last decade (climate already changed) and not to the histori-

 Both *acclimation* of individual plants during their lifetime and adaptation of a plant population (*evolutionary process*) can contribute to the vegetation's stomatal conductance response to temperature (e.g., Juan et al.; 2011). The evolutionary process would lead to a decrease in the number of individuals of those species whose natural acclimation range differs strongly from the new norm. As "perennial grassland" in our model consists of a range of species, the grassland as a whole may rapidly adapt.

In addition, there is the phenomenon of *epigenetics*, exemplified by Norway spruce trees grown from seeds. The temperatures experienced during seed maturation (climate at the site where the mother tree stands) affect the new trees' response to environmental factors determining bud set in autumn. This means that the seedlings may be adapted to the site where they are produced, although the genetic background of the parents predicts a different response (e.g. Skrøppa et al.; 2010).

- Line 298. "We construct cold as representative for a species that is more tolerant to cold temperatures, but slightly less efficient at warm temperatures compared to MM. This is accomplished by moving $T_{opt}$ towards cooler temperatures while keeping the other parameters fixed to MM values". From Figure 6a and Table 1 I see that for the "cold" parameterization not only $T_{opt}$ was moved, but also $T_{min}$ for (e.g. for grassland). Indeed, we will correct the text.

- Figure 7. The $g_{stom}/g_{max}$ ratio in the subarctic parameterization with PPFD0.8 is greater in the morning than at noon. How then the choice of PPFD08 is explained? Please comment on this in the text. PPFD0.8 was chosen to maximize the relative stomatal conductance at noon and in the morning. Please refer to L326–334. We will elaborate on the text and make it clearer.

- Line 334. I don't understand how we can say that the differences are "substantial". I don't see much difference between deciduous trees (a) and grassland (c), sorry. There seems to be a misunderstanding? The referee refers to the sentence "As expected due to the small adjustments, coniferous trees display the smallest differences between the different parameterizations, while the differences for perennial grassland are substantial as a response to the proposed temperature acclimation." Figure 7 is meant to be read for each species separately and from right to left. It comprises the mean and standard deviation of relative stomatal conductance computed from hourly meteorological data over 21 years. Panel (c) displays perennial grassland. Furthest to the right, the results using the mapping manual parameterization are shown. The relative stomatal conductance around noon when the highest values are expected is rather low (about 40%). With our adjusted parameterization, the relative stomatal conductance around noon reaches about 70–80%. The standard deviation reflects the variability of growing conditions and should not be interpreted as a statistical measure to separate one distribution from the other. Even more, data would perhaps not reduce the standard deviation in this

case. We will remove the term "substantial" and make our interpretation of the Figure clearer in the text.

- Line 336. Using net photosynthesis to calculate leaf emergence is not completely justified. Leaves are likely to be present and active well before gross photosynthesis equals heterotrophic respiration (eg. soil respiration). Gross photosynthesis should be used to calculate Astart and Aend instead. Thank you for pointing this out. It is correct that net photosynthesis does not capture real photosynthesis. However, gross photosynthesis is not among the available products from MODIS satellites. Instead, we use gross primary production which is including maintenance respiration. According to the product website (MOD17, last accessed Dec. 2021) it is computed from observed absorbed photosynthetically active radiation (APAR), a species-specific conversion efficiency parameter $\epsilon(\text{temperature}, \text{waterstress})$, temperature, and water stress. In Fig. 1 the estimated start and end of the growing season from MODIS (Aqua/Terra) GPP over a $1 \times 1$ km patch centered at Svanhovd is shown. They differ only slightly from the estimates using net photosynthesis. The start of the growing season is shifted by one day (later) and the end by one/two days (later). Considering the temporal resolution of 8 days of the satellite product this difference is negligible.

[Figure]

Figure 1: Estimated start and end of growing season from MODIS (Aqua/Terra) GPP over a $1 \times 1$ km patch centered at Svanhovd. The average start in 2018 amounts to doy 123 and to 107 in 2019. The end of the growing season in 2018 is doy 262 and in 2019 276.

- Line 354. "A sample of downy birch leaves collected at Svanhovd had an average length of (3.0±0.5)cm" Were top-canopy leaves sampled? How many leaves were collected to get +- 0.5 cm standard error? The birch (B. pubescence) leaves were collected from the outer canopy (good light exposure) of tall trees at two sites in Finnmark. The leaves were collected late in July so they were fully expanded. They were collected by hand, so at about 2 m height. As the sun is at a low elevation angle for most of the time in the growing season in this area, light exposure at the top of the tree is not expected to be so different from lower leaves as long as they grow in a part of the tree that is not shadowed by other trees. At the first site (Karasjok), three leaves were collected from each of five adult trees. At the second site (Svanvik), five leaves were collected from one tree. The leaves were pressed and dried before scanning for the area and shape determination through image analysis. ImageJ was used for thresholding the images to give silhouettes of the leaves. After scaling the images, the program was used for finding the minimum Feret diameter. This measure finds the largest width of a leaf at a $90°$ angle to the length of the leaf (the Feret diameter). With this method we found that the 20 leaves had a mean width of 3.014 cm and a standard deviation of 0.4996 cm, given as $(3.0 \pm 0.5)$ cm in the manuscript.

- Line 355. "We used 13.5m height" Why was this value chosen? What is the meaning of a height between the average tree height and the maximum tree heigh? Perhaps it would have been more reasonable to use the average height. We suppose, the referee refers to the 10.1 m average height of one particular Scots pine forest which has been measured in 2004. We assumed that after 14 years 10.1 m might not be the average height of this particular forest anymore. In the absence of more recent measurements, we found it more reasonable to choose 13.5 m the average tree height in the whole area in 2004 which included all tree species and ages. We will include this in the text.

- Line 360 and following. POD1 was calculated by gap filling the data, right? Because there is a lot of data missing in the middle of the season. Or were POD1 compensated for missing data? If so, how? Please confirm it by writing it in the text. In Appendix B2, we write that the "DO$_3$SE model requires hourly, continuous meteorological observations." This means that all input data, including ozone, have to be gap-filled before POD1 can be computed. The gap-filling method for all data but ozone is described in appendix B2. The used gap-filling method for ozone has been summarized above (response to "Lines 206–208") and published in Falk et al. (2021). We shall restructure the text to make the description of the DO$_3$SE model and its input data clearer.

- Line 369. "Due to the shape of flight, a symmetric variation". Symmetric variation of what? This refers to the variation of PPFD at a stomatal opening of 50% by $\pm 20\%$ and the resulting response in computed POD1. We will make this clearer in the text.

- Line 370. "We find that an opening of stomata at lower light intensities can cause higher sensitivity to drought conditions." Please, explain where we can see this. Graph 8 is not clear at all to me. And then, "sensitivity" of what? Of plants? Of POD1? Figure 8 is indeed very complex and we will elaborate on the explanation and interpretation in the text to make it more accessible to the readers. You are right that this sentence is not clear and does not communicate what we intended to say. We refer to the sensitivity of modeled POD1. If we take a look at, e.g. Fig. 8a subarctic parameterization and 2018 (left-hand side of the upper panel). Open symbols represent a model simulation where SWP was taken into account. Closed symbols where this effect was switched off. The '−' symbolizes PPFD0.8 (earlier opening), the '+' PPFD1.2 (later opening). We shall have a look at the circles connected by a solid line representing the simulations with a GS which we have determined from temperatures. If the stomata open earlier / close later more ozone can be taken up (longer opening time), hence POD1 is larger than for the unchanged $f_{\text{light}}$. Now, if we take SWP into account, we find that only the PPFD0.8 is affected. Therefore, we conclude that conditions that negatively affect SWP (referred to as droughts) reduce the uptake of ozone (POD1) when an earlier opening / later closing of stomata is considered. Hence, POD1 is more sensitive to drought conditions.

- Line 373. "The magnitude of these effects varies between PFTs as well as years, but the predicted ozone uptake for the bespoke temperature parameterization is always larger than for the MM parameterizations and of the same order of magnitude as the variability between the years studied here." What effects? "Of the same order of magnitude as the interannual variability...": can you conclude it by comparing only two years? The same for line 402 In Section 3, we tried to show that 2019 is representative of a normal year while 2018 is more extreme. We should evaluate our data with this question in mind and confirm that the difference between 2018 and 2019 is indeed representative of the interannual variability. Depending on the outcome, we will rephrase the sentences and substitute the term "interannual variability".

- Table 4. Have the percentage of reduction been calculated taking into account pre-industrial concentrations as prescribed by the MM? What are meaning of the superscripts? And, above all, why some superscripts indicate a range (e.g. 1.9 ... 2.1)? I did not understand how the stdev of the MM estimation was calculated, sorry. We will make this clearer in the next version of the manuscript. We used the relationship between biomass reduction and POD1 unchanged from the MM. This, however, is to some degree questionable because subarctic vegetation could be affected more or less by the same ozone dose as the central European species. The caption of the table is not complete and parts of the explanation missing. We explain how we derive the uncertainty ranges in L405–407 but this may not be clear enough. From Fig. 8, we deduced that SWP is negligible in most cases (solid and open symbols show the same POD1). Hence, we computed the biomass reduction based on simulations with SWP effects on POD1 turned off and MM $f_{\text{light}}$. The uncertainties reported are the differences and not standard deviations. The differences are calculated between the simulation with MM $f_{\text{light}}$ and "bespoke" GS and the simulations with PPFD0.8 (larger POD1 $\rightarrow$ larger biomass reduction) and PPFD1.2 (smaller POD1 $\rightarrow$ less biomass reduction). The sub- and super-scripts show the asymmetry introduced by the different $f_{\text{light}}$ parameterizations. The range represents the additional uncertainty from the choice in GS.

- Line 416. "we have developed bespoke parameterizations" it seems a bit strong statement to me, you have not developed any new tailored parameterization, you have only hypothesized one. There is no one experiment nor comparison with experimental results in your work. We will substitute the term to appropriately reflect our intentions and work.

- Line 417. "The comparison between meteorological conditions in 2018 and 2019 and their divergence from climatology allowed us to assess the influence of key environmental variables such as temperature, PPFD, and precipitation on vegetation susceptibility to O3 damage in light of future changes as may occur under climate change" I did not understand where all this "divergence with the climatological average" of these two years alone lies, sorry. We will make this clearer in the revision of the manuscript.

- Line 432. "With respect to ongoing climate change, a clear positive trend emerged in length (5.2d decade-1) of the growing season that is almost equally distributed between earlier start (2.9 days decade-1) and later end (2.3d decade-1) (Appendix Fig. A1)." How did you figure it out? Have you been doing retrospective MODIS analysis for 30 years? Or do you have a publication to quote? We have derived the number from an analysis of the thermal start and end of the GS based on the gridded temperature data from SeNorge.no provided by the Norwegian weather service and shown in Appendix Fig. 9. We may remove this part eventually or rephrase the sentence to include this information, support it with citations of relevant works, and include this analysis in a methods section.

- Line 435 and following. "visible damage" Visible damage and POD can be totally unrelated, as demonstrated by some research conducted on agricultural species. I recommend caution in stating that the O3 peaks causing the visible symptoms can result in a biomass reduction (damage). You are right. We intended to state that visible damage and biomass reduction are not necessarily related in this paragraph. We agree that this is not clear and we will improve on this.

- Line 441. Does "damage" mean "visible leaf symptoms"? Or does it mean biomass reduction? See above. We will carefully rephrase the text concerning the term "damage" and clearly distinguish between visible damage and biomass reduction.

- Line 456. "We found that soil water potential under 2018/19 meteorological conditions was negligible" What does it mean? That there was no water in the soil

(SWP were negligible) or that the effect on the POD of the presence or absence of SWP in the calculation was negligible? Please clarify. The latter is correct. Thank you for pointing out this ambiguity.

- Line 461. "better suited" Point 1 is questionable. Also point 2 is questionable. How can you say that the MM parameterization does not capture the plant physiology of subactic vegetation if no comparisons with physiological measurements taken on subarctic vegetation are presented? We will address this point by reframing the storyline of the manuscript towards a new method to derive stomatal conductance parameterizations based on climatological and remotely sensed data.

- Line 469. "However, the decline of this ozone spring peak is partly caused by the uptake of vegetation" Are you sure? Please cite a reference. We will rephrase this sentence to "[...] could partly be caused [...]". And may give a recap of ozone removal, as follows: *ozone is removed from the atmosphere by dry deposition which is described as a network of resistances. In general, snow and ice surfaces are found to have a low resistance to ozone, hence less ozone can be taken up by the land surface which is a major sink in absence of photochemical reactions (winter). Therefore, existing ozone accumulates in the subarctic winter boundary layer. The same goes for non-biogenic precursors. In spring these are chemically reactivated and increase ozone production. At the same time, coniferous trees might also start to produce BVOCs. At the same time, most of the vegetation is still covered by snow and ice and dry deposition remains low. In addition to the intrusion of stratospheric ozone, this causes the enhancement of ozone at the surface (spring peak). Now, with snowmelt and bud burst in spring, ozone dry deposition increases. Simultaneously, photochemical destruction of ozone also increases. Only part of ozone enters the stomata.*

- Line 491. "Automation of the here proposed PDF-based acclimation using machine learning techniques could overcome these issues in the future" What does it mean? Please explain. Make an example. The proposed method to use climate data for adjusting the parameterization of stomatal conductance can be formulated as an optimization problem (maximizing the enclosed temperature PDF area) which would make it possible to derive new parameters in a more automized way. Now, considering enough stomatal conductance data were available from observation in well-known climate conditions these data could be used to evaluate our method and train a model to find the optimal stomatal conductance parameterization based on climate data alone. As climate data is more readily available compared to the actual measurement of stomatal conductance this could help to improve modeling of stomatal conductance globally.

- Figure A1. How was the length of the growing seasons in the various years identified? By satellite? Other method? What does the gray band represent? See response to "Line 432". The gray band represents the standard deviation. We will either remove or elaborate on this analysis.

- Line 511. "with fmin, Dmin, Dmax describing the relative stomatal conductance to changes in vapor pressure deficit." It is not clear. Please, clarify what D and fmin are, and their units.

-
    - Line 517. "The DO3SE model as described in Büker et al. (2012) is used to simulate SWP0 across a PFT specific root depth according to the Penman–Monteith energy balance method that drives water cycling through the soil–plant–atmosphere system" I cannot understand how the P-M energy balance is used in DO3SE to derive the SWP. Please explain in detail. This is described in Bücker et al (2012) in detail. We will include an improved, short description of how the DO$_3$SE model incorporates the PM model to estimate evapotranspiration and hence water loss from the soil. However, for full details of this method, readers should refer to Bücker et al. (2012).

    - Line 525. "the concentration at the upper surface of the laminar layer for a sunlit upper canopy leaf" At what height was the O3 concentration measured? If it was not measured at the top of the canopy (10m for trees or 10 cm for grassland), how was the O3 concentration at the top canopy calculated? Please explain in detail. The DO$_3$SE model was used to estimate the difference in [O$_3$] between a reference height above the canopy and the canopy height. This employs the deposition component of the DO$_3$SE model that estimates the transfer of mass (i.e. ozone) as a function of wind speed, convection, surface roughness, and vegetation [O$_3$] sink. We will add additional detail in the paper to make this clear.

    - Line 526. What does rc represent? Is it the cuticular resistance or the bulk canopy resistance? What is its value? This represents cuticular resistance - we will include its value in the revision of the paper.

    - Line 528. Can you explain where that formula for calculating the flux comes from? Why is there u(z1) in? And what is the z1 height? This comes from the DO$_3$SE model - we will add a suitable reference.

    - Line 531. What is the z1 height? Where is it? See above.

    - Line 535. Wind speed at 2 m: what is it used for? Please explain See above.

    - Section B1. The description of fPHEN is missing. Please, provide it. Again, how do you calculate the day-to-day SWP on your site? Please describe it in detail. See above.

  In summary, we will make the description of the DO$_3$SE model more comprehensible and comprehensive. However, in the interest of the main focus of this manuscript, it may not be possible to give a full recap of all details concerning this well-established model.

- Line 550. Please explicitly describe the method used to gap-fill O3 concentrations because it could be crucial. See response to "Lines 206–208" above.

- Line 554. "From Fig. B1f) it is apparent that the mapping manual parameterized grassland would not have been able to grow in 2019." It does not seem to me that gstom has been reseted at all. If this is the case, the premises of the work appear weak. As mentioned earlier, in the revision of our work, we will emphasize that the main objective of our work is to explore the possibilities of readily accessible data (i.e. long-term meteorological temperature observations, $CO_2$ flux estimations from remote sensing) to provide alternatives to site-specific data that is needed to parameterize plant physiology in models such as the $DO_3SE$ model used in our study.